# M-FAC: Efficient Matrix-Free Approximations of Second-Order Information

**Elias Frantar**
IST Austria
elias.frantar@ist.ac.at

**Eldar Kurtic**
IST Austria
eldar.kurtic@ist.ac.at

**Dan Alistarh**
IST Austria & Neural Magic
dan.alistarh@ist.ac.at

## Abstract

Efficiently approximating local curvature information of the loss function is a key tool for optimization and compression of deep neural networks. Yet, most existing methods to approximate second-order information have high computational or storage costs, which limits their practicality. In this work, we investigate matrix-free, linear-time approaches for estimating Inverse-Hessian Vector Products (IHVPs) for the case when the Hessian can be approximated as a sum of rank-one matrices, as in the classic approximation of the Hessian by the empirical Fisher matrix. We propose two new algorithms: the first is tailored towards network compression and can compute the IHVP for dimension $d$, if the Hessian is given as a sum of $m$ rank-one matrices, using $O(dm^2)$ precomputation, $O(dm)$ cost for computing the IHVP, and query cost $O(m)$ for any single element of the inverse Hessian. The second algorithm targets an optimization setting, where we wish to compute the product between the inverse Hessian, estimated over a sliding window of optimization steps, and a given gradient direction, as required for preconditioned SGD. We give an algorithm with cost $O(dm + m^2)$ for computing the IHVP and $O(dm + m^3)$ for adding or removing any gradient from the sliding window. These two algorithms yield state-of-the-art results for network pruning and optimization with lower computational overhead relative to existing second-order methods. Implementations are available at [9] and [17].

## 1 Introduction

Given the recent success and increasing impact of deep learning, there has been significant work on improving the fundamental technical tools underpinning its progress. One such tool is the ability to estimate the local geometry of the loss function for deep models, which often comes in the form of estimates for second-order (Hessian) information. Such information is critical in several settings, such as neural network optimization and pruning.

Directly using Hessian information in the context of deep learning is infeasible: for example, just storing the Hessian matrix for the standard ResNet50 model [14] would occupy 2.5 Petabytes. These constraints have inspired significant work on efficient numerical approximations of the Hessian for deep neural networks, such as the line of work on the K-FAC approximation [19, 31, 5, 49], or efficient block-wise approximations [47, 37]. One of the classic approaches, which we focus on in this paper, is the *empirical Fisher* [13, 3, 4] approximation to the Hessian, written as:

$$\mathbf{H} \simeq \widehat{\mathbf{F}} = \frac{1}{N} \sum_{i=1}^{N} \nabla \ell_i \cdot \nabla \ell_i^{\top}, \tag{1}$$

where $N$ is the number of samples, $\nabla \ell_i$ denotes the gradient w.r.t. the $i$th sample at the given point, and $\cdot$ is the outer product of individual gradients, which we view as column vectors.

The empirical Fisher approximation is fairly standard, e.g. [13, 3, 4], and has been recognized to be useful in a variety of practical settings where exact estimation of the Hessian or of the true Fisher information matrix is not feasible. At the same time, there is still active research in the community

on the conditions for its applicability [29, 21, 37, 40]. A useful property of this approximation is that it also allows to estimate the *inverse* of the Hessian, which is essential in many applications.

Specifically, the fact that the empirical Fisher can be written as a sum of rank-one matrices allows the use of the Woodbury-Sherman-Morrison inversion formula [46] to exactly compute its inverse, by recursively integrating terms corresponding to each gradient into the inverse. (Please see Equation 2 for an exact derivation.) This approach was independently proposed by [13, 3], for pruning and optimization, respectively, where it was validated on small networks, with hundreds of weights.

The idea was adapted to deep neural networks (DNNs) by [37], through approximation of the inverse in *small diagonal blocks*. The authors show improved approximation quality for the Hessian inverse relative to a simple diagonal approximation, and that this leads to state-of-the-art pruning results in terms of accuracy. Yet, this approach is limited by the block-wise approximation: for block size $B$ and dimension $d$, it requires $\Theta(Bdm)$ time to recursively build the block-wise Fisher approximation using $m$ gradients, and $\Theta(Bd)$ time and memory for computing the Inverse-Hessian-Vector-Products (IHVPs) necessary for estimating pruning statistics. Clearly, the ideal $B = d$ case is still intractable at scale, and generally it is still unknown whether efficient algorithms are possible for computing IHVPs in this context.

**Contribution.** We address this question by introducing two efficient algorithms for computing IHVPs under the empirical Fisher approximation, with computational and storage costs that are *linear* in the dimension $d$ of the model, without the need for block-wise approximations, assuming that the number of samples $m$ in the approximation is constant. Concretely, we provide exact *matrix-free* algorithms to compute products of the form $\widehat{\mathbf{F}}^{-1}\mathbf{v}$, where $\widehat{\mathbf{F}}^{-1}$ is the inverse empirical Fisher, and $\mathbf{v}$ is an arbitrary vector. We show that these algorithms can be implemented efficiently, and that they can match or improve the state-of-the-art results for both neural network pruning and optimization.

**The Static Algorithm.** Our first algorithm assumes a *static* scenario, which is standard in neural network pruning: we are given a fully-trained model $\theta^\star$, for which we wish to estimate IHVPs and diagonal elements of the inverse Hessian, in order to determine the "optimal" pruning update using e.g. the Optimal Brain Surgeon (OBS) framework [24, 13]. For this, we first compute $m$ gradients at $\theta^\star$, which we will use to estimate IHVPs via the empirical Fisher and compute pruning statistics.

The main idea is that, since we only wish to compute products between the inverse and an arbitrary vector (IHVPs), we can rewrite the Woodbury recursion such that we work *exclusively with individual vectors and scalars*, and never with full $d \times d$ or $d \times B$ matrices. Given model dimension $d$ and $m$ gradients, each defining a rank-one component of the empirical Fisher, the algorithm uses $O(dm^2)$ pre-computation time, and will have $O(dm)$ cost for *exactly* computing the IHVP. Further, we can specialize the algorithm to directly query elements of the Hessian inverse, at a cost of $O(m)$ time per element. This provides efficient, linear-in-$d$ implementations for all operations required by the OBS pruning framework. Finally, we note that the static algorithm can also be applied in a block-wise manner without any change in the total compute and memory costs.

**The Dynamic Algorithm.** Our main contribution is in extending this idea to preconditioned SGD optimization, i.e. to precondition stochastic gradients by our estimate of the inverse Hessian. We start from the classic idea of bootstrapping the approximation by leveraging previous gradients: the preconditioner at time $t$ is built from gradients obtained during a "sliding window" over the last $\Delta \geq 1$ optimization steps. This requires a *dynamic* representation, allowing addition and removal of gradients *without full recomputation* of second-order statistics.

We show that this can be achieved, with approximately $O(dm)$ time and space complexity. The key idea is that, for any ordered set $(\nabla \ell_j)_{j=1}^m$ of $m$ gradients, and any vector $\mathbf{v}$, it is possible to represent the corresponding IHVP estimate $\widehat{\mathbf{F}}_m^{-1}\mathbf{v}$ as a linear combination of terms corresponding to individual gradients $\nabla \ell_j$ and $\mathbf{v}$, of the form $\widehat{\mathbf{F}}_m^{-1}\mathbf{v} = \lambda^{-1}\mathbf{v} - \sum_{j=1}^m c_j^m \nabla \ell_j$, where $\lambda > 0$ is a dampening constant.

Crucially, we ensure that the coefficients $c_j^m$ can be computed just via dot products $\nabla \ell_i^\top \nabla \ell_j$, where $i \leq j$ in the ordering, and $\nabla \ell_j^\top \mathbf{v}$. Then, to replace a given gradient $\nabla \ell_i$ from this representation, we just have to compute $m$ scalar products with the new gradient vector (as well as update some intermediate information). Hence, the entire update operation has computational cost $O(dm + m^3)$ for replacing any gradient in the sliding window, and $O(dm + m^2)$ for computing the IHVP.

**Implementation and Experiments.** We provide efficient vectorized implementations for the above algorithms, called *M-FAC*, for *Matrix-Free Approximate Curvature*. Specifically, M-FAC consists of

Pytorch [34] implementations of a pruning and optimization library. Our implementation introduces several additional optimizations, in particular GPU acceleration via custom CUDA kernels, and minimizes the cost of memory transfer between the GPU and main memory via memory paging.

For pruning, our implementation provides order-of-magnitude improvements over the block-wise approximation of [37] for classic benchmarks such as pruning ResNet50 and MobileNet on the ImageNet dataset. This allows us to obtain more accurate sparse models by exploring higher parameter settings and increasing the total number of pruning steps, while remaining practical in terms of memory and compute even for larger models. What is more, our preconditioned SGD (even without momentum) can be competitive in terms of validation accuracy with state-of-the-art optimizers on models of moderate size, including compact vision architectures and Transformer language models [42]. Its computational overheads are of 5%–55% relative to vanilla SGD on standard CNN architectures.

## 2    Preliminaries and Related Work

**General Definitions.**    We now briefly introduce the setting and notation; we refer the reader to standard texts [29, 27, 4] for a complete introduction. We start from the standard setting in which we are given a dataset $\mathcal{D} = (\mathbf{x}_i, \mathbf{y}_i)_{i=1}^N$, and wish to identify a $d$-dimensional model $\theta \in \mathbb{R}^d$ to minimize an empirical loss $L : \mathbb{R}^d \to \mathbb{R}$, defined as $L(\theta) = \frac{1}{N} \sum_{i=1}^N \ell(\mathbf{y}_i, f(\mathbf{x}_i; \theta))$. For a twice-differentiable loss $L$, the Hessian is the matrix of second-order derivatives of $L$ w.r.t. $\theta$, i.e. $\mathbf{H} = \nabla_\theta^2 L$. In the *probabilistic* view, each input example $\mathbf{x}$ has some probability of being assigned a given label $\mathbf{y}$. Given input examples $\mathbf{x}$ drawn from a distribution $Q_{\mathbf{x}}$ and corresponding outputs drawn from a conditional distribution $Q_{\mathbf{y}|\mathbf{x}}$, the goal is to minimize the distance between the target joint distribution $Q_{\mathbf{x},\mathbf{y}} = Q_{\mathbf{x}} Q_{\mathbf{y}|\mathbf{x}}$, and a learned joint distribution $P_{\mathbf{x},\mathbf{y}}(\theta)$, where $\theta$ is the model.

**The Fisher Matrix.**    Assuming the probabilistic view, it can be shown that the Fisher information matrix $\mathbf{F}$ of the model's joint distribution satisfies $\mathbf{F} = \mathrm{E}_{P_{\mathbf{x},\mathbf{y}}} \left[ -\nabla_\theta^2 \log p_{\mathbf{x},\mathbf{y}}(\theta) \right]$ where $p_{\mathbf{x},\mathbf{y}}(\theta)$ is the density function. If the model's output conditional distribution matches the conditional distribution of the data, then the Fisher and Hessian matrices are in fact equivalent [27]. Roughly, this means that, if the model has high accuracy, we can approximate the Hessian of $L$ at $\theta$ with the Fisher matrix.

It is sometimes useful to consider an approximation to the Fisher, where the distribution $P_{\mathbf{x},\mathbf{y}}$ is replaced with the empirical training distribution $\widehat{Q}_{\mathbf{x},\mathbf{y}}$, leading to the *empirical Fisher matrix*:

$$\widehat{\mathbf{F}} = \mathrm{E}_{\widehat{Q}_{\mathbf{x}}} \left[ \mathrm{E}_{\widehat{Q}_{\mathbf{y}|\mathbf{x}}} \left[ \nabla \log p_{\mathbf{y}|\mathbf{x}}(\theta) \nabla \log p_{\mathbf{y}|\mathbf{x}}(\theta)^\top \right] \right] = \frac{1}{N} \sum_{i=1}^N \underbrace{\nabla \ell(\mathbf{y}_i, f(\mathbf{x}_i; \theta))}_{\nabla \ell_i} \nabla \ell(\mathbf{y}_i, f(\mathbf{x}_i; \theta))^\top.$$

**The Woodbury-Sherman-Morrison Trick.**    The fact that this approximation is a sum of rank-one matrices has the benefit that it allows efficient exact computation of the Fisher inverse. Specifically, one can apply the classic Woodbury-Sherman-Morrison formula to compute the inverse recursively as

$$\widehat{\mathbf{F}}^{-1} = \widehat{F}_N^{-1} = \widehat{F}_{N-1}^{-1} - \frac{\widehat{F}_{N-1}^{-1} \nabla \ell_N (\nabla \ell_N^\top \widehat{F}_{N-1}^{-1})}{N + \nabla \ell_N^\top \widehat{F}_{N-1}^{-1} \nabla \ell_N}, \tag{2}$$

where the recursion is over samples, and the base step is $\widehat{F}_0^{-1} = \lambda^{-1} I_d$, with $\lambda$ being a small positive constant. This approach was independently introduced by Hassibi and Stork [13] for pruning, and Amari [3] for natural gradient. Singh and Alistarh [37] showed that it can be scaled to DNNs by block-wise approximation, and that it provides better approximations of the loss than K-FAC-based [44], or diagonal [39] approximations [7], leading to more accurate pruning. The main shortcoming of directly applying (2) is the prohibitive computational cost of $\Omega(d^2 m)$, even when reduced to $\Omega(dBm)$ by $B$-block-wise approximation. Our method proposes a *matrix-free* approach for exactly calculating IHVPs and querying individual elements of the inverse Hessian, of cost $O(dm)$ after $O(dm^2)$ precomputation. Our method is numerically equivalent to the direct computation above.

**Diagonal Approximations.**    A common approximation, both for optimization, e.g. [20, 18] but also for pruning [39], is to assume that the Fisher matrix is *diagonal*. In our notation, this method has setup cost $O(dm)$, and diagonal query cost $O(d)$. However, as evident from the experimental results of [37] this provides lower approximation quality relative to both block-wise methods or K-FAC [44].

**K-FAC Approximations.**    This approach observes that the entries of the true Fisher corresponding to blocks "between" two layers, which can be written as the expectation of a Kronecker product

between two matrices, can be approximated as the Kronecker product of the expectations of those two matrices (reversing the order between the product and the expectation). This approximation has been leveraged for both pruning and for optimization, and it allows efficient computation of the inverse [5, 33, 49, 44, 23]; however, it is known to not always hold [31]. Another relative drawback is that the Kronecker factorization only occurs naturally for fully-connected layers; but there is work on extensions to other layer types [11, 30], via additional approximations. We compare against K-FAC-based optimizers and pruners, and find that our method yields better results.

**Additional Approaches.** Our approach is similar in spirit to matrix-free methods [32, 25, 28], but the algorithms we present are new. *Hessian-free* optimization [28] also forgoes the explicit computation of Hessians in favor of computing an IHVP with a vector $\mathbf{v}$. However, this estimation is performed by iteratively approximating the solution to the linear system $\mathbf{Hx} = \mathbf{v}$ for some given $\mathbf{x}$ without ever explicitly forming $\mathbf{H}$. One disadvantage of this method in practice is that it requires tuning and several very costly iterations to converge (for a single $\mathbf{v}$), as the underlying Hessian can be ill-conditioned. The L-OBS pruning method [7] approximates second-order information by defining independent layer-wise objectives, which allows the direct approximation of the layer-wise Hessians via a carefully-crafted block structure. In contrast, our approach allows for fully-global Hessian estimation and it also yields better pruning results at scale.

*Full-matrix adaptive regularization* has similar goals, but in the context of adaptive optimizers [8]. Agarwal et al. [2] proposed GGT, which allows the efficient computation of the *inverse square root* of the low-rank matrix resulting from the sum of gradient outer products over a sliding window. At its core, this procedure requires an eigen-decomposition (implemented via SVD) of an $m \times m$ matrix at every time step, which is reasonably efficient for small values of $m$.

Our dynamic algorithm solves a similar, but slightly simpler problem, as we only want to invert the matrix, without computing its square root. We do not perform any eigen-decompositions; instead, we carefully maintain intermediate information that allows an efficient explicit computation of the scalar coefficients in Equation 6. At the end of Section 3.2, we discuss how this approach is more efficient in practice and allows executing at larger window sizes (with small overhead), which leads to improved model accuracy as we show in our experiments. Additionally, our dynamic algorithm has per-step cost $O(dm + m^3)$, relative to $O(dm^2 + m^3)$ reported by GGT [2]; this can result in lower overhead versus GGT, even when the SVD cost is negligible (e.g. if $m$ is small).

Yao et al. [47] recently provided an alternative method for approximating the diagonal of the inverse Hessian, using Hutchinson's randomized algorithm for estimating the diagonal. To mitigate the variance, the authors introduce non-trivial smoothing heuristics. Their approach has theoretical per-iteration cost of at least $2\times$ versus SGD, as it requires a second backward pass over the network; and this cost is usually higher in practice. Experimentally, our algorithm often matches their accuracy, and, unlike AdaHessian, its cost depends exclusively on the model size, irrespective of the underlying structure. Thus, as we show in the experimental section, our average *practical overhead* is less than 50% for dense models, and less than 10% for sparse ones.

**Approximation Quality.** Künstner et al. [21] performed an in-depth analysis of the empirical Fisher, making the point that, in theory, the approximation could become meaningless if all sample gradients are zero. Similarly to [37], we did not find that this occurs in practice for deep neural networks, as sample gradients are never zero. The latter reference provides detailed comparisons of diagonal, K-FAC, and other approximations, and finds that the empirical Fisher can provide competitive approximation quality for DNNs. Further, they demonstrate that better loss approximation implies better accuracy for neural network pruning. We therefore do not repeat their loss analysis, and mainly compare methods in terms of application performance.

## 3 Algorithm Descriptions

### 3.1 The Static Algorithm

As a warm-up, we first describe the IHVP algorithm given a static set of gradients. While our notation is customized for the empirical Fisher, all our techniques are applicable to any matrix that can be written as the sum of $m$ rank-one components. Specifically, we are given $m$ vectors, which we assume to be the gradient vectors $(\nabla \ell_i)_{i=1}^m$, and must compute quantities related to the inverse of the matrix resulting from the sum (more precisely, the average) of their outer products, which we denote by $\widehat{F}_m^{-1}$. The main idea is to rewrite the recursive description of $\widehat{F}_m^{-1}$ such that, after some precomputation, the matrix-vector-product $\widehat{F}_m^{-1}\mathbf{x}$ can be computed efficiently for any vector $\mathbf{x}$. Using

the Sherman-Morrison formula applied to the partial empirical Fisher matrix $\widehat{F}_i$ corresponding to the first $i$ gradients (and scaled by $1/m$), we obtain the following recursion:

$$\widehat{F}_i^{-1}\mathbf{x} = \widehat{F}_{i-1}^{-1}\mathbf{x} - \frac{\widehat{F}_{i-1}^{-1}\nabla\ell_i(\widehat{F}_{i-1}^{-1}\nabla\ell_i)^\top}{m + \nabla\ell_i^\top \widehat{F}_{i-1}^{-1}\nabla\ell_i}\mathbf{x}, \text{ with } \widehat{F}_0^{-1}\mathbf{x} = \lambda^{-1}I_d\mathbf{x} = \lambda^{-1}\mathbf{x} \text{ and } \lambda > 0. \quad (3)$$

For simplicity, let us set $\mathbf{v_i} = \widehat{F}_{i-1}^{-1}\nabla\ell_i$ and unroll the Sherman-Morrison recursion, which gives:

$$\widehat{F}_i^{-1}\mathbf{x} = \widehat{F}_{i-1}^{-1}\mathbf{x} - \mathbf{v_i}\frac{\mathbf{v_i}^\top\mathbf{x}}{m + \nabla\ell_i^\top\mathbf{v_i}} = \lambda^{-1}\mathbf{x} - \sum_{j=1}^{i}\mathbf{v_j}\frac{\mathbf{v_j}^\top\mathbf{x}}{m + \nabla\ell_j^\top\mathbf{v_j}}. \quad (4)$$

**IHVP Computation.** Assuming that we have already computed all the vectors $\mathbf{v_j}$, the above expression can be calculated in $O(dm)$ time without any intermediate $d \times d$ matrices: first compute the scalar fractions in (4), and then evaluate the resulting linear combination of $\mathbf{x}$ and $\mathbf{v_j}$. Since $\mathbf{v_i} = \widehat{F}_{i-1}^{-1}\nabla\ell_i$, it can be computed in exactly the same way given all $\mathbf{v_j}$ for $j < i$. Thus, all $m$ vectors $\mathbf{v_i}$ of dimension $d$ can be precomputed in increasing order of $i$ using $O(dm^2)$ total time. As an additional optimization (to cut the required memory in half), we also precompute all $q_j = m + \nabla\ell_j^\top\mathbf{v_j}$.

**Querying the Inverse.** To extract individual elements $[\widehat{F}_m^{-1}]_{ij}$ of the inverse Hessian approximation, we can write $\mathbf{e_i}^\top\widehat{F}_m^{-1}\mathbf{e_j}$ in the form of (4) where $\mathbf{e_i}$ and $\mathbf{e_j}$ are indicator vectors

$$\mathbf{e_i}^\top\widehat{F}_m^{-1}\mathbf{e_j} = \mathbf{e_i}^\top\lambda^{-1}\mathbf{e_j} - \sum_{k=1}^{m}\mathbf{e_i}^\top\mathbf{v_k}\frac{\mathbf{v_k}^\top\mathbf{e_j}}{q_k}. \quad (5)$$

As $\mathbf{e_i}^\top\mathbf{v_k}$ and $\mathbf{v_k}^\top\mathbf{e_j}$ can both be realized as constant time indexing operations, the above turns into a sum over $m$ scalars. Hence, our method admits $O(m)$ access to any element of $\widehat{F}_m^{-1}$ using the same precomputed $\mathbf{v_k}$ and $q_k$ as for the efficient calculation of $\widehat{F}_m^{-1}\mathbf{x}$.

**Additional Optimizations.** The algorithm admits a fast vectorized implementation, and several optimizations, which we describe in the Appendix. For example, we perform several memory-saving optimizations, as well as explicit page swapping between CPU and GPU memory to mitigate the gradient transfer costs. Furthermore, the static algorithm can be applied independently to each block of a block-wise approximation: for block size $B$, the computation and memory costs per block are reduced by $B/d$, but since there are now $d/B$ blocks, the overall costs will stay the same irrespective of $B$. Thus, as long as $m < B$, our method is $B/m$ times faster (and less memory intense) than the direct implementation of the Woodbury inverse.

## 3.2 The Dynamic Algorithm

We now describe the *dynamic* algorithm, which assumes that gradients arrive in an online fashion and must be integrated into the Fisher estimate. We first present the algorithm itself, i.e. the setup / update / IHVP computations, and perform a complexity analysis. Next, we show how the algorithm can be derived and finally we conclude with notes on an efficient practical implementation.

**High-Level View.** The main idea of the dynamic algorithm is to write the IHVP as

$$\widehat{F}_m^{-1}\mathbf{x} = \lambda^{-1}\mathbf{x} - \sum_{j=1}^{m}c_j^m\nabla\ell_j, \quad (6)$$

where the scalar coefficients $c_j^m$ can be computed efficiently from just the scalar products $\nabla\ell_i^\top\nabla\ell_j$ and $\nabla\ell_i^\top\mathbf{x}$. Then, any gradient $\nabla\ell_i$ can be replaced by updating just $O(m)$ of the stored scalar product values. In the following, we use $\mathbf{G} = [\nabla\ell_1^\top; \nabla\ell_2^\top; \ldots; \nabla\ell_m^\top]$ to denote the row-wise $m \times d$ matrix of the $m$ gradients for which we wish to compute the inverse empirical Fisher. Further, for each $i$ from 1 to $m$, let $\mathbf{b^i}$ be a vector with $m$ components $(b_1^i, b_2^i, \ldots, b_m^i)$, such that

$$\widehat{F}_{i-1}^{-1}\nabla\ell_i = \lambda^{-1}\nabla\ell_i - \sum_{j=1}^{i-1}a_j^{i-1}\nabla\ell_j = \sum_{j=1}^{m}b_j^i\nabla\ell_j, \quad (7)$$

i.e. containing the scalar coefficients of $\widehat{F}_{i-1}^{-1}\nabla\ell_i$. This means that $b_j^i = -a_j^{i-1}$ when $j < i$, $b_i^i = \lambda^{-1}$, and $b_j^i = 0$ otherwise.

**Initial Precomputation & Update.** The dynamic algorithm maintains three $m \times m$ matrices: $\mathbf{GG}^\top$, $\mathbf{D}$ and $\mathbf{B}$. The first is the symmetric gradient scalar product matrix $[\mathbf{GG}^\top]_{ij} = \nabla\ell_i^\top\nabla\ell_j$.

The second is an upper triangular matrix and stores the precomputed values $[\mathbf{D}]_{ij} = \nabla \ell_i^\top \widehat{F}_{i-1}^{-1} \nabla \ell_j$ for $i \leq j$. The third is the row-wise matrix of the coefficient vectors $[\mathbf{B}]_{ij} = b_j^i$, which makes it lower triangular with a diagonal of $\lambda^{-1}$. We now discuss how to compute those matrices.

The initial setup of the dynamic algorithm begins by evaluating $\mathbf{G}\mathbf{G}^\top$ in a straightforward fashion. Next, $\mathbf{D}$ can be computed for $i \leq j$ according to the following recursion:

$$[\mathbf{D}]_{ij} = (\nabla \ell_i^\top \widehat{F}_{i-1}^{-1} \nabla \ell_j) \text{ and } (\nabla \ell_j^\top \widehat{F}_0^{-1} \nabla \ell_k) = \lambda^{-1}[\mathbf{G}\mathbf{G}^\top]_{jk} \qquad (8)$$

$$(\nabla \ell_j^\top \widehat{F}_i^{-1} \nabla \ell_k) = (\nabla \ell_j^\top \widehat{F}_{i-1}^{-1} \nabla \ell_k) - \frac{(\nabla \ell_j^\top \widehat{F}_{i-1}^{-1} \nabla \ell_i)(\nabla \ell_i^\top \widehat{F}_{i-1}^{-1} \nabla \ell_k)}{m + (\nabla \ell_i^\top \widehat{F}_{i-1}^{-1} \nabla \ell_i)}. \qquad (9)$$

Given $\mathbf{D}$, we can then conclude the precomputation by calculating $\mathbf{B}$ for $i \geq j$ recursively as:

$$[\mathbf{B}]_{ii} = \lambda^{-1} \text{ and } [\mathbf{B}]_{ij} = -\sum_{k=j}^{i-1} \frac{[\mathbf{D}]_{ki}}{m + [\mathbf{D}]_{kk}} [\mathbf{B}]_{kj}. \qquad (10)$$

After the initial setup, gradient $\nabla \ell_k$ can be replaced with gradient $\nabla \ell'_k$ by first updating row $k$ of $\mathbf{G}$ and then replacing row and column $k$ in $\mathbf{G}\mathbf{G}^\top$ with $\mathbf{G}\nabla \ell'_k$. Afterwards, the recomputation of columns $j \geq k$ of $\mathbf{D}$ and rows $i \geq k$ of $\mathbf{B}$ completes the update.

**Multiplication.** Once $\mathbf{G}\mathbf{G}^\top$, $\mathbf{D}$ and $\mathbf{B}$ have been precomputed, one can perform efficient IHVPs of the form $\widehat{F}_m^{-1}\mathbf{x}$ with arbitrary vectors $\mathbf{x}$. This is done by first evaluating $\mathbf{p} = \mathbf{G}\mathbf{x}$ and then computing all $m$ values $q_i$ by the following recursion:

$$q_i = \frac{(\nabla \ell_i^\top \widehat{F}_{i-1}^{-1}\mathbf{x})}{m + [\mathbf{D}]_{ii}} \text{ and } (\nabla \ell_j^\top \widehat{F}_0^{-1}\mathbf{x}) = \lambda^{-1}p_j \qquad (11)$$

$$(\nabla \ell_j^\top \widehat{F}_i^{-1}\mathbf{x}) = (\nabla \ell_j^\top \widehat{F}_{i-1}^{-1}\mathbf{x}) - \frac{[\mathbf{D}]_{ij}}{m + [\mathbf{D}]_{ii}}(\nabla \ell_i^\top \widehat{F}_{i-1}^{-1}\mathbf{x}). \qquad (12)$$

Eventually, the final result of the IHVP is obtained by:

$$\widehat{F}_m^{-1}\mathbf{x} = \lambda^{-1}\mathbf{x} - \sum_{j=1}^{m}(\sum_{k=j}^{m} q_k[\mathbf{B}]_{kj})\nabla \ell_j. \qquad (13)$$

**Complexity Analysis.** The dynamic algorithm stores $m$ gradients of dimension $d$ as well as three $m \times m$ matrices, and thus has an overall memory complexity of $O(dm + m^2)$.

Next, we analyze the time complexity of all important operations. Initially, $\mathbf{G}\mathbf{G}^\top$ must be computed once, which takes $O(dm^2)$ time. Then, the recursion of $\mathbf{D}$ has three indices with $1 \leq i, j, k \leq m$ and each step takes constant time. Thus, it can be computed in $O(m^3)$ time with dynamic programming. Further, since values for index $i$ depend only on values for index $i - 1$, the dynamic programming can be implemented in $O(m^2)$ space. $\mathbf{B}$ has two indices $1 \leq i, j \leq m$ and every recursion takes $O(m)$ time, meaning that it can also be fully computed in $O(m^3)$ through dynamic programming. Hence, the overall initial setup cost is $O(dm^2 + m^3)$.

To replace one gradient with $\nabla \ell'$, we have to compute $\mathbf{G}\nabla \ell'$ as well as (partially) recalculate $\mathbf{D}$ and $\mathbf{B}$, which takes at worst $O(dm + m^3)$ time. An IHVP requires two matrix-vector products involving $\mathbf{G}$ and a recursion with two indices and therefore has a complexity of $O(dm + m^2)$.

**Algorithmic Derivation.** The dynamic algorithm can be derived directly from Theorem 1, which we state here in a simplified form (using the definitions of $\mathbf{B}$ and $\mathbf{D}$), and prove in the Appendix.

**Theorem 1.** *Let* $\widehat{F}_i = \lambda I + 1/m \cdot \sum_{j=1}^{i} \nabla \ell_j \nabla \ell_j^\top$, *then* $\widehat{F}_i^{-1}\mathbf{x}$ *can be calculated as:*

$$\widehat{F}_i^{-1}\mathbf{x} = \lambda^{-1}\mathbf{x} - \sum_{j=1}^{i} c_j^i \nabla \ell_j \text{ and } c_j^i = \sum_{k=j}^{i} \frac{(\nabla \ell_k^\top \widehat{F}_{k-1}^{-1}\mathbf{x})}{m + [\mathbf{D}]_{kk}}[\mathbf{B}]_{kj}. \qquad (14)$$

Equation (14) with $i = m$ is exactly equal to the IHVP computation (13) as the fraction in the innermost sum corresponds to $q_k$. Similarly, index shifting $i = i - 1$ and setting $x = \nabla \ell_i$, thus turning $c_j^i$ into $-[\mathbf{B}]_{ij}$ as well as the fraction's numerator to $[\mathbf{D}]_{ki}$, recovers the precomputation formula of $\mathbf{B}$ given by (10) for $i > j$. The formulas for $\mathbf{D}$ and $q_i$ follow directly from an expansion of the Woodbury formula followed by an appropriate recursive evaluation that avoids any matrix / vector operations except in the base case (we indicate the corresponding recursive calls by brackets).

**Efficient Implementation.** Directly implementing the discussed recursive formulas in a modern machine learning framework would result in very slow code. Fortunately, it is possible to implement all computations required for the dynamic algorithm very efficiently on a GPU. We describe how to do this in the Supplementary Material and provide a full implementation [9]. Specifically, we provide complete sample code that is able to perform the calculation of $\mathbf{D}$ and $\mathbf{B}$, i.e. the $O(m^3)$ component of the overall update cost, in $< 10$ milliseconds (on an NVIDIA RTX 2080 Ti) for values of $m$ as high as 1024 (and the code can still be further optimized). For reference, this is $> 10\times$ faster than the highly-optimized $m \times m$ SVD computation done at every step by GGT [2]. We emphasize that this $O(m^3)$ computation being very fast in practice is crucial to reach low overheads, especially when dealing with a models where the $O(dm)$ matrix-vector products are not the bottleneck.

## 4 Experimental Validation

### 4.1 Application 1: Pruning DNNs using the Static Algorithm

**Background.** Given a trained model $\theta^\star$, the goal of pruning is to find the weight $\theta_i$, or the set of weights, whose setting to $0$ would lead to a minimal increase in the training loss. Under a local quadratic approximation of the loss function, the OBD framework [24] shows that the "optimal" weight to be removed is the one with the lowest value of the saliency metric $\rho(\theta_i) = \frac{\theta_i^2}{2\,[\widehat{\mathbf{F}}^{-1}]_{ii}}$ and proposed to estimate $[\widehat{\mathbf{F}}^{-1}]_{ii}$ by diagonal approximation. The OBS framework [13] observed that the *remaining* weights $\theta$ should also be updated via the optimal perturbation $\delta\theta = -\theta_i \widehat{\mathbf{F}}^{-1} \mathbf{e}_i / [\widehat{\mathbf{F}}^{-1}]_{ii}$, where $\mathbf{e}_i$ is the $i$th basis vector. Our algorithm efficiently supports both these operations.

Wang et al. [44] raised the valid point that applying the OBS update above to multiple weights being removed at once may be incorrect, since it ignores possible correlations between those weights. We considered this point in detail, comparing results between OBD pruning [24], the OBS update [13], and an augmented version of OBS which disentangles the correlations by solving a corresponding linear system [37]. The results, presented in the Appendix, suggest that the OBS update is quite beneficial even in its approximate form, and that the effect of correlation is small for unstructured pruning. In fact, we find that the OBS pruning mask in each step is usually very similar to the one obtained by simple magnitude pruning, suggesting that the final accuracy improvements are primarily due to the updates of the remaining weights, facilitating better recovery.

**Experimental Setup.** We prune CNNs (ResNet-50 [15] and MobileNet-V1 [16]) on the ImageNet dataset [36]. These models are standard in the pruning literature [10], and therefore several strong baselines exist. Timing experiments are run on a machine with NVIDIA RTX 2080 Ti GPUs, a 48-core Intel CPU, and 512 GB of RAM. Following [37], we used batched gradients (of size 16) as single samples inside the Fisher approximation. This does not alter results, but reduces variance.

We compare against Global Magnitude Pruning (GMP) [12], Layer-wise Optimal Brain Surgeon (L-OBS) [7], Soft Threshold Reparametrization (STR) [22], and WoodFisher (WF) [37]. The latter two methods are state-of-the-art for gradual pruning. WF is numerically equivalent to our method at the same parameter settings, but has significantly higher computational and storage cost. Relative to it, we therefore focus on executing at higher parameter values. We compare against their public implementation. The Appendix contains full hyper-parameters, ablations with respect to block size and number of gradients, and a comparison with K-FAC pruning in a simpler setting [49].

We first perform a one-shot comparison to evaluate the "raw" per step pruning performance of M-FAC relative to other methods. Next, we evaluate how simply increasing M-FAC parameter values (but keeping everything else exactly the same as in WF's experiments [37] which follow [50]) improves over state-of-the-art gradual pruning results by WF, GMP and STR. Finally, we demonstrate how M-FAC's high per-step efficiency can be utilized to craft practical pruning schedules with little computational overhead relative to GMP and only moderate extra memory consumption, which at the same time yield significantly better results than WF's state-of-the-art numbers.

**Oneshot Pruning.** Figure 1a shows the Top-5 accuracy of one-shot ResNet50 pruned models for L-OBS, GMP, WoodFisher, and M-FAC, where our method is executed with a parameter setting that is infeasible for WoodFisher due to the required computational and storage costs. (We use Top-5 accuracy to compare with [7].) We note the improved accuracy of global methods relative to layer-wise, and that the Fisher approximation yields consistently better results relative to global magnitude. This suggests that a better approximation of the Fisher inverse also yields better pruning results, and is in line with the loss approximation study of [37].

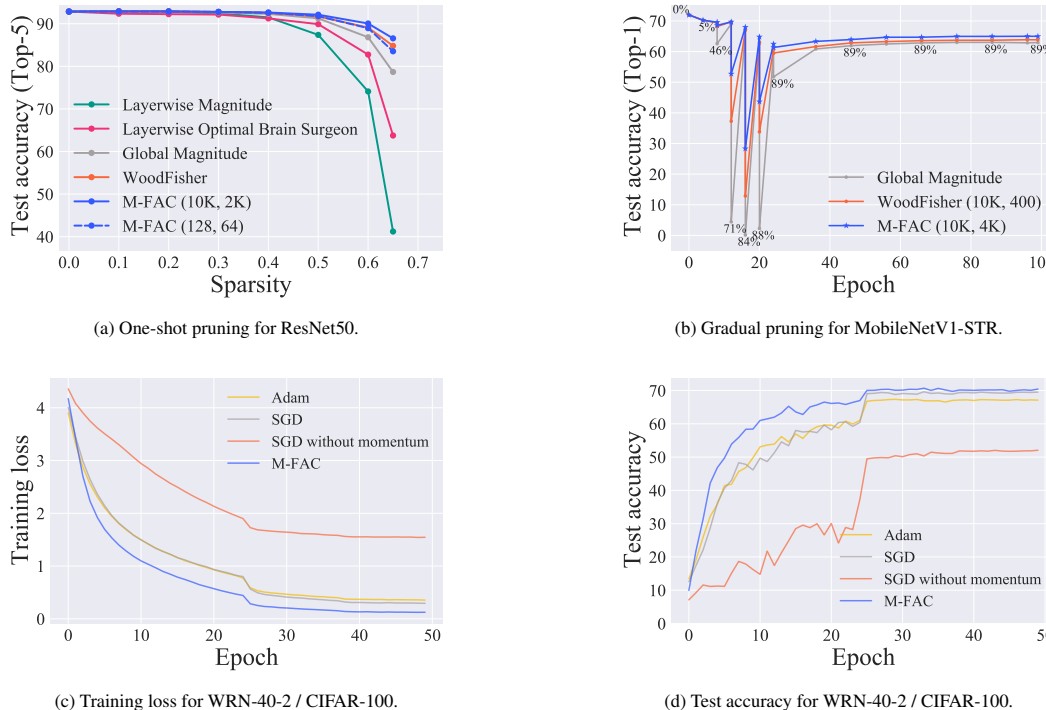

(a) One-shot pruning for ResNet50.

(b) Gradual pruning for MobileNetV1-STR.

(c) Training loss for WRN-40-2 / CIFAR-100.

(d) Test accuracy for WRN-40-2 / CIFAR-100.

Figure 1: **(a)** Accuracy vs. sparsity for one-shot pruning on ResNet50 / ImageNet. **(b)** Accuracy during gradual pruning for the MobileNetV1-STR / ImageNet experiment. **(c)** and **(d)** Comparison of different optimizers on a short WRN-40-2 / CIFAR-100 schedule.

Generally, we found that estimation using more gradients always improves results, until saturation. Interestingly, increasing the block size does not necessarily improve results, for some models, smaller block sizes sometimes yield better results for a fixed number of gradients.

Figure 1b examines the effect of the improvement in one-shot pruning accuracy at each step of pruning, on the final result, when pruning MobileNetV1-STR gradually to $89\%$ sparsity. WoodFisher uses block size $10K$, and $m = 400$ gradients, while M-FAC uses the same block size but $m = 4000$ gradients. Executing WoodFisher with the same parameters would be extremely slow. Note the gap in one-shot accuracy following each pruning step: this translates in the final accuracy gap of more than $1\%$ even after the extensive fine-tuning phase.

**Gradual Pruning Comparisons.** Table 1 presents gradual pruning results for MobileNetV1-STR/ImageNet at 89% sparsity and ResNet50/ImageNet at 95% sparsity, relative to other state-of-the-art methods. M-FAC outperforms previous methods in the case of MobileNetV1-STR by more than 1.5% Top-1 accuracy. Further, despite using $10\times$ more gradients to estimate the empirical Fisher, the per-step pruning cost of M-FAC is $> 6\times$ lower than WoodFisher's as can be seen in Table 2 (for the same parameters settings M-FAC is $> 100\times$ faster). The ResNet50 results show smaller gains.

Table 1: Gradual pruning results (top-1 accuracy) for **MobileNetV1-STR / ImageNet** at 89% sparsity and **ResNet50 / ImageNet** at 95% sparsity.

| Method | MBv1 Top-1 (%) | RN50 Top-1 (%) |
|---|---|---|
| Dense baseline | 72.00 | 77.01 |
| STR [22] | 62.10 | 70.40 |
| Global Magnitude | 62.94 | 71.72 |
| WoodFisher [37] | 63.87 | 72.12 |
| **M-FAC** | **65.01** | **72.32** |

Table 2: Minutes per pruning step compared with WoodFisher (WF). OOM denotes out of memory on a system with 512GB RAM.

| Model | $B$ | $m$ | WF | **M-FAC** |
|---|---|---|---|---|
| MBv1-STR | 10k | 400 | 60m | **0.5m** |
| MBv1-STR | 10k | 4k | OOM | **8.7m** |
| MBv1-STR | all | 1k | OOM | **1.2m** |
| ResNet50 | 2k | 400 | 35m | **2.5m** |
| ResNet50 | 10k | 1k | OOM | **14.0m** |
| ResNet50 | all | 512 | OOM | **4.7m** |

**Practical Pruning.** Due to the computational cost of a WF step, the previous schedules used only a small number of pruning steps. However, M-FAC makes it possible to prune much more frequently. In this case, the cost of extracting and storing several thousand gradients may still be a bottleneck.

We find that M-FAC already performs well with low parameter settings, especially when coupled with recomputations (i.e. resampling gradients and rebuilding the M-FAC approximation after partial pruning steps). We now perform MobileNet and ResNet50 gradual pruning runs using M-FAC with block size $B = 128$, 64 gradients and 16 recomputations per step. A single M-FAC pass takes less than a few seconds at these settings. Details about the pruning schedule can be found in the Appendix. This is a practical setup, which can scale to larger models. At the same time, Table 3 shows that leads to significantly improved pruning results, sometimes even by several percent accuracy, compared to WF and the M-FAC results in Table 1.

|  | MBv1-0% | MBv1-75% | MBv1-89% | RN50-0% | RN50-95% | RN50-98% |
|---|---|---|---|---|---|---|
| WoodFisher | 72.0 | 70.1 | 63.9 | 77.0 | 72.1 | 65.6 |
| **M-FAC** | 72.0 | **71.0** | **67.3** | 77.0 | **72.6** | **67.5** |

Table 3: M-FAC performance with optimized practical settings.

## 4.2 Application 2: Optimization using the Dynamic Algorithm

**Matrix-Free Preconditioned SGD.** The dynamic algorithm can provide an efficient implementation for the following variant of SGD. Let $t$ be the current iteration index, and $\Delta < t$ be the length of the sliding window during which the estimation of the inverse Hessian is performed. Consider the iteration $\theta_{t+1} = \theta_t - \eta_t \widehat{F}_{[t-\Delta,t]}^{-1} \nabla \ell_t$, where $\eta_t$ is the learning rate. Let $\nabla \ell_{t-\tau+1}, \ldots, \nabla \ell_t$ be the gradients obtained at steps $(t - \tau + 1), \ldots, t$, respectively. Then, we define the preconditioning matrix as $\widehat{F}_{[t-\Delta,t]} = \lambda^{-1} I_d + 1/\Delta \cdot \sum_{\tau=0}^{\Delta-1} \nabla \ell_{t-\tau} \cdot \nabla \ell_{t-\tau}^\top$, that is, an empirical Fisher matrix generated with respect to the sliding window of $\Delta$ gradients. At each step, our dynamic algorithm replaces $\nabla \ell_{t-\Delta+1}$ with $\nabla \ell_t$ in the inverse estimate, and then computes $\widehat{F}_{[t-\Delta,t]}^{-1} \nabla \ell_t$. This corresponds to full-matrix natural gradient SGD [3] under the extra assumption that gradients do not change too quickly during training. This assumption has been validated in the K-FAC literature [5] and we also provide additional experimental justification in the Appendix. Finally, to better study the effect of the preconditionier, we do *not* apply momentum to M-FAC in any of the experiments below.

**Initial Test.** We begin by verifying that the M-FAC optimizer actually behaves as one would expect from a second-order method, i.e. learning faster than standard first-order techniques, especially in the early epochs. For that purpose, we run M-FAC, SGD with and without momentum and Adam on WideResNet 40-2 / CIFAR10 with a compressed 50 epoch schedule. The corresponding Figures 1c and 1d indeed showcase the expected behavior. Interestingly, they also show that SGD without momentum performs poorly in this setup, which further highlights the major impact of the M-FAC preconditioner, which seems to be able to make up for the lack of momentum.

**Second Order Comparison.** Next, we work with the common ResNet20 and ResNet32 models [15] and compare against standard optimizers such as SGD with momentum, Adam [18] and AdamW [26], but also against approximate second-order methods such as K-FAC [30], GGT [2], and AdaHessian [47]. For the former two, we use the implementations contributed to TensorFlow [1] by the authors, while for the latter we use the authors' PyTorch implementation. For fairness, we always follow the settings recommended by the authors, although this makes overheads harder to compare: we used specific learning rates, and grid-searched the weight-decay values for each method in turn.

Since implementations have slightly different framework requirements, our main performance metric is *overhead over SGD*, measured in each algorithm's environment, on an NVIDIA RTX 3090 GPU for PyTorch and a Titan RTX for TensorFlow. For all second-order methods, we compute the preconditioner at each step—reducing the frequency reduces overheads proportionally for each method, but introduces an additional hyper-parameter and generally results in lower final accuracy. We provide additional information in the Supplementary.

Our first results are presented in Table 4 (left), where we examine the best Top-1 test accuracy obtained by each method, as well as its overhead relative to SGD. For K-FAC, the best accuracy is obtained with batch size 1000, while all other methods operate on batch size 128. Similarly, GGT's best results are achieved with window size $m = 100$, while for M-FAC larger window sizes improve results; we use $m = 512$. Due to these discrepancies, we measure overheads with the "best accuracy" setting for each method but also in the same setup as M-FAC; the latter is given in brackets.

| | ResNet20 | | ResNet32 | | | Model | SGD | Adam | M-FAC (Overh.) |
|---|---|---|---|---|---|---|---|---|---|
| Method | Acc. | Overhead* | Acc. | Overhead* | | | | | |
| SGD | 91.78 | 1.00× | 92.80 | 1.00× | | WRN 22-2 | **69.93** | 66.90 | 69.76 (1.40×) |
| Adam | 89.67 | 1.05× | 90.56 | 1.10× | | WRN 40-2 | 71.75 | 70.14 | **72.42** (1.35×) |
| AdamW | 91.78 | 1.05× | 92.58 | 1.10× | | WRN 22-4 | 73.13 | 72.52 | **74.06** (1.43×) |
| | | | | | | MBv1 Dense | 68.06 | 67.92 | **68.96** (1.25×) |
| K-FAC | 91.65 | 1.21× (2.35×) | 90.09 | 1.50× (3.05×) | | | | | |
| GGT | 88.38 | 1.05× (8.35×) | 89.14 | 1.05× (8.88×) | | MBv1 Sparse | 64.11 | – | **64.78** (1.03×) |
| AdaHessian | 92.17 | 3.50× | **92.81** | 3.30× | | RN50 Sparse | 74.78 | – | **75.10** (1.05×) |
| M-FAC | **92.34** | 1.55× | 92.65 | 1.50× | | | | | |

Table 4: **Left**: Comparison of M-FAC against first- and second-order optimizers with individual tuning and weight decay. **Right**: Additional experiments with no tuning and weight decay. Sparse results are from finetuning after the last step of gradual pruning to $\approx 90\%$ sparsity. * For K-FAC / GGT we indicate in parentheses the overhead in a comparable setting to M-FAC with the same batchsize / the same number of gradients.

| | | SQv2 | SST-2 | MRPC | STS-B | QQP | MNLI-m | MNLI-mm | QNLI |
|---|---|---|---|---|---|---|---|---|---|
| t | Adam | 48.41 | 80.11 | 69.90 | 64.39 | 81.09 | 65.36 | 67.78 | 77.85 |
| t | M-FAC | **49.80** | **81.86** | **72.94** | **80.15** | **84.20** | **68.28** | **68.98** | **81.17** |
| m | Adam | 54.80 | **85.46** | 76.57 | 82.09 | 86.45 | 73.30 | 74.85 | **83.85** |
| m | M-FAC | **58.02** | 84.20 | **78.87** | **84.66** | **86.75** | **74.59** | **75.95** | 83.70 |

Table 5: Average of 5 runs performance of Adam and M-FAC for tiny (t) and mini (m) BERT on SQuADv2 and GLUE benchmark tasks. Due to space, we only show accuracy for SQuADv2 / QQP and the Pearson correlation for STS-B, see Appendix for full results. Models are hosted at `https://huggingface.co/M-FAC`.

In terms of accuracy, M-FAC scores highest in the ResNet20 experiment, and comes third, to tuned SGD and AdaHessian (performing almost the same), on ResNet32. The overhead of M-FAC is around $50\%$, which is lower than AdaHessian, on par with K-FAC, but higher than GGT. However, these overheads are comparable only when GGT uses $5\times$ less gradients (surprisingly, larger window sizes yield worse results for GGT). On the same settings, M-FAC provides up to 5x lower overhead. This is due to (a) better performance of our dynamic algorithm relative to SVD and (b) the $\Theta(dm^2)$ term in GGT, versus our $O(dm)$ term.

**Image Classification.** Table 4 (right) provides a scenario where we run SGD with momentum, Adam, and M-FAC *without tuning or weight-decay* on Wide Residual Networks (WRN) [48] for CIFAR-100, as well as on MobileNetV1/ ImageNet, and examine test accuracy. M-FAC achieves the highest accuracy on most runs, even on ImageNet. We emphasize that these results can probably be improved by parameter tuning. Another interesting application is fine-tuning sparse models. We fine-tune 90% sparse MobileNet / ResNet50 models for 30 epochs starting from the last gradual pruning step and find that M-FAC reaches higher final accuracy than SGD in both cases. At the same time, it is only negligibly slower ($\leq 5\%$ overhead) since algorithm's complexity (compute and memory) is linear in the model *density*. At last, we note that M-FAC achieving higher test accuracy is generally well correlated with a lower final training loss.

**Language Modelling using Transformers.** Finally, we test the M-FAC optimizer for smaller Transformer models. We use the default values $m = 1024$ gradients (unless the dataset has less than this number of total samples), dampening $\lambda = 10^{-6}$, learning rate $10^{-4}$ and no weight decay. We then train BERT [6] *tiny* and *mini* models [41] on SQuADv2 [35] and the GLUE benchmark suite [43], comparing against HuggingFace's [45] Adam baseline. For M-FAC, we use exactly the same training setup and only replace the optimizer. The results in Table 5 show that M-FAC performs better than the Adam baseline on almost all tasks, on several even by considerable margins. We provide more detailed results in the Appendix, including hyperparameters and a comparison with AdamW.

## 5 Discussion

We presented static and dynamic algorithms for computing IHVPs when the Hessian matrix can be approximated by a sum of rank-1 matrices. We used the classic empirical Fisher approximation, but our results can apply more broadly. The main limitation is the cost of storing $m$ additional gradients. For the static algorithm, we can efficiently leverage system memory, as described in the Appendix. The dynamic algorithm could parallelize gradient storage, or perform gradient compression [38]. We plan to investigate this in future work and perform a larger-scale study of optimization performance.

# 6 Acknowledgements

We gratefully acknowledge funding the European Research Council (ERC) under the European Union's Horizon 2020 research and innovation programme (grant agreement No 805223 ScaleML), as well as computational support from Amazon Web Services (AWS) EC2.

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
