# Appendix

# Additional Material

## A Proof of Theorem 1

We now prove the Theorem 1, which forms the basis for the dynamic algorithm. Before doing that, we restate the theorem in its full version:

**Theorem.** *Let $(\nabla \ell_i)_{i=1}^m$ be a series of gradients, and, for any index $i \geq 1$, let $\widehat{F}_i = \lambda I_d + \frac{1}{m} \sum_{j=1}^i \nabla \ell_j \cdot \nabla \ell_j^\top$ be a dampened version of the empirical Fisher, where $\lambda > 0$ is a small constant. Then $\widehat{F}_i^{-1} \mathbf{x}$ can be calculated as:*

$$\widehat{F}_i^{-1} \mathbf{x} = \lambda^{-1} \mathbf{x} - \sum_{j=1}^i c_j^i \nabla \ell_j \ \ and \ \ c_j^i = \sum_{k=j}^i \frac{\left(\nabla \ell_k^\top \widehat{F}_{k-1}^{-1} \mathbf{x}\right)}{m + [\mathbf{D}]_{kk}} [\mathbf{B}]_{kj}, \tag{15}$$

*where $[\mathbf{D}]_{kk} = \nabla \ell_k^\top \widehat{F}_{k-1}^{-1} \nabla \ell_k$, and $[\mathbf{B}]_{kj}$ being defined such that*

$$\widehat{F}_{k-1}^{-1} \nabla \ell_k = \sum_{j=1}^k [\mathbf{B}]_{kj} \nabla \ell_j \ \ and \ \ [\mathbf{B}]_{11} = \lambda^{-1}. \tag{16}$$

*Proof.* The proof makes use of the following two equalities:

$$\widehat{F}_i^{-1} \mathbf{x} = \lambda^{-1} \mathbf{x} - \sum_{j=1}^i c_j^i \nabla \ell_j \tag{17}$$

$$\widehat{F}_i^{-1} \mathbf{x} = \lambda^{-1} \mathbf{x} - \sum_{j=1}^i \left(\widehat{F}_{j-1}^{-1} \nabla \ell_j\right) \frac{\left(\widehat{F}_{j-1}^{-1} \nabla \ell_j\right)^\top \mathbf{x}}{m + [\mathbf{D}]_{jj}}, \tag{18}$$

which correspond to Equation 6 with $m = i$ and Equation 3 in an unrolled form, respectively. We begin by setting them equal. Next, we apply (17) again to $\widehat{F}_{j-1}^{-1} \nabla \ell_j$ (naming the corresponding coefficients $a_k^{j-1}$) and reorganize the nested sums to get a new expression of the form (17). Finally, we simplify the result by using our definition of $[\mathbf{B}]_{ij} = -a_j^{i-1}$ for $j < i$ and $[\mathbf{B}]_{ii} = \lambda^{-1}$ (discussed at the beginning of Section 3.2). We obtain:

$$\widehat{F}_i^{-1} \mathbf{x} = \lambda^{-1} \mathbf{x} - \sum_{j=1}^i c_j^i \nabla \ell_j \tag{19}$$

$$= \lambda^{-1} \mathbf{x} - \sum_{j=1}^i \left(\widehat{F}_{j-1}^{-1} \nabla \ell_j\right) \frac{\left(\widehat{F}_{j-1}^{-1} \nabla \ell_j\right)^\top \mathbf{x}}{m + [\mathbf{D}]_{jj}} \tag{20}$$

$$= \lambda^{-1} \mathbf{x} - \sum_{j=1}^i \left(\lambda^{-1} \nabla \ell_j - \sum_{k=1}^{j-1} a_k^{j-1} \nabla \ell_k\right) \frac{\left(\nabla \ell_j^\top \widehat{F}_{j-1}^{-1} \mathbf{x}\right)}{m + [\mathbf{D}]_{jj}} \tag{21}$$

$$= \lambda^{-1} \mathbf{x} - \sum_{j=1}^i \left(\frac{\left(\nabla \ell_j^\top \widehat{F}_{j-1}^{-1} \mathbf{x}\right)}{m + [\mathbf{D}]_{jj}} \lambda^{-1} - \sum_{k=j+1}^i \frac{\left(\nabla \ell_k^\top \widehat{F}_{k-1}^{-1} \mathbf{x}\right)}{m + [\mathbf{D}]_{kk}} a_j^{k-1}\right) \nabla \ell_j \tag{22}$$

$$= \lambda^{-1} \mathbf{x} - \sum_{j=1}^i \left(\sum_{k=j}^i \frac{\left(\nabla \ell_k^\top \widehat{F}_{k-1}^{-1} \mathbf{x}\right)}{m + [\mathbf{D}]_{kk}} [\mathbf{B}]_{kj}\right) \nabla \ell_j. \tag{23}$$

We can now directly match coefficients between (19) and (23), as they hold for any gradient values. This completes the proof. $\square$

## B Efficiently Implementing the Dynamic Algorithm

As mentioned in Section 3.2, directly implementing the various recursive formulas of the dynamic algorithm will most likely be quite slow in practice. Thus, we now discuss how to develop an efficient

practical implementation. The reader can also find the full code for this implementation of the dynamic algorithm (and the corresponding optimizer) in PyTorch, available at [11].

**Setup.** We begin by vectorizing the calculations of $\mathbf{D}$ and $\mathbf{B}$. Algorithm 1 describes the full setup procedure in PyTorch-like pseudocode.

---

**Algorithm 1** Calculating $\mathbf{D}$ and $\mathbf{B}$ in $O(m^3)$ time and $O(m^2)$ memory, assuming that $\mathbf{GG}^\top$ is already precomputed.

---

$\mathbf{D} \leftarrow \lambda^{-1}\mathbf{GG}^\top$
$\mathbf{B} \leftarrow \lambda^{-1}\mathbf{I}_{m \times m}$

**for** $i \leftarrow 2, 3, \ldots, m$ **do**
$\quad \mathbf{D}_{i:,i:} \leftarrow \mathbf{D}_{i:,i:} - \frac{1}{m + D_{i-1,i-1}} \mathbf{D}_{i-1:,i:}^\top \mathbf{D}_{i-1:,i:}$
**end for**
$\mathbf{D} \leftarrow \mathrm{triu}(\mathbf{D})$

**for** $i \leftarrow 2, 3, \ldots, m$ **do**
$\quad \mathbf{B}_{i,:i} \leftarrow (-\mathbf{D}_{:i,i}/(m + \mathrm{diag}(\mathbf{D})_{:i}))^\top \mathbf{B}_{:i,:i}$
**end for**

---

Initially, $\mathbf{D}$ is the stored scalar product matrix $\mathbf{GG}^\top$ scaled by $\lambda^{-1}$. Row 1 already has the correct final values. The algorithm then goes through $m - 1$ iterations, each completing one row but also updating all the rows of higher index. More concretely, in iteration $i$, we subtract the outer product of the previous row starting at the element one after the diagonal $\mathbf{D}_{i-1,i:}$ with itself, scaled by the inverse of $m$ plus the previous diagonal element $\mathbf{D}_{i-1,i-1}$, from the square sub-matrix with $ii$ as the upper left corner, i.e. $\mathbf{D}_{i:,i:}$. This process is also visualized in Figure 2, where the already calculated elements are shaded in light blue, the ones that are currently being updated in darker blue, and the irrelevant ones in grey. Since the calculation of $\mathbf{D}$, as given here, will also produce some unnecessary (and incorrect) values below the diagonal, we delete those after the loop by extracting just the upper triangular part of $\mathbf{D}$. $\mathbf{B}$ starts off as the identity matrix times $\lambda^{-1}$ (and row 1 is again already done). Next, $\mathbf{B}$ is calculated row by row, where row $i$ is a negative linear combination of the previous $i - 1$ rows (up to index $i - 1$ as $\mathbf{B}$ is lower triangular) $\mathbf{B}_{:i,:i}$ with the coefficients given by the first $i - 1$ elements of column $i$ of $\mathbf{D}$, i.e. $\mathbf{D}_{:i,i}$, each divided by $m$ plus the corresponding diagonal element of $\mathbf{D}$, i.e. $m + \mathrm{diag}(\mathbf{D})_i$. This process is also shown in Figure 2.

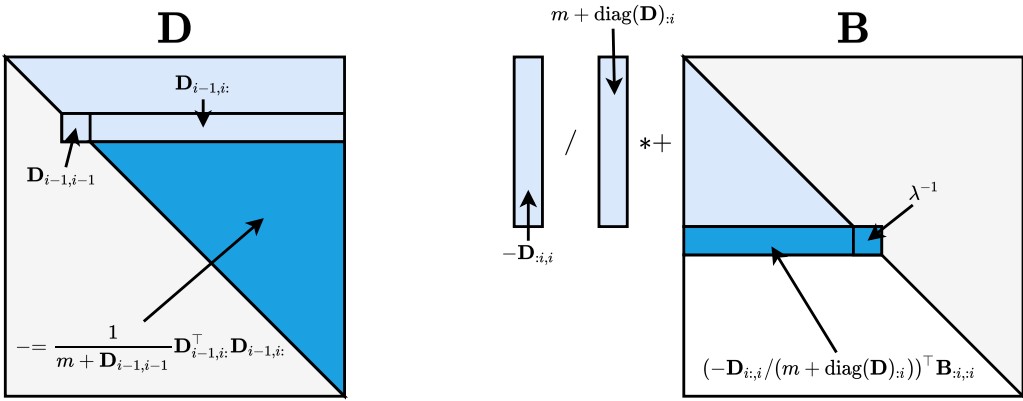

Figure 2: Visualized calculation of $\mathbf{D}$ and $\mathbf{B}$.

In theory, the algorithm presented so far should be well suited for utilizing the massive parallel computation capabilities of a GPU. However, a straightforward implementation in e.g. PyTorch, will not be very fast for larger values of $m$ as it results in thousands of small kernel launches with overhead that adds up to a very significant overall slow-down. (Concretely, we measured $> 7\times$ overhead for $m = 1024$). This means that, to achieve good performance, the setup for the coefficient

computation of the dynamic algorithm needs to be implemented with (custom) CUDA kernels that merge computations to get rid of the excessive kernel launch overhead.

Carefully studying the calculation process of $\mathbf{D}$ reveals that it is actually very similar to a Cholesky decomposition (concretely, the only difference is that it does not use the square root of the diagonal element but the element itself plus $m$). In fact, it is actually exactly equivalent to taking the upper triangular matrix in the LU-decomposition (with no pivoting) of $\lambda^{-1}\mathbf{G}\mathbf{G}^\top + m\mathbf{I}$ and then subtracting $m\mathbf{I}$ again. This means that it is possible to reuse the highly optimized LU-decomposition kernel provided by PyTorch to calculate $\mathbf{D}$ quite efficiently (a custom kernel implementing the modified Cholesky decomposition would certainly be faster, but we already observed pretty low overheads with the LU version). For the efficient computation of $\mathbf{B}$, a custom kernel is needed, which we provide at [11], and describe next.

The main idea is that we can dramatically reduce the number of separate kernel launches by computing multiple rows of $\mathbf{B}$ in a single kernel. In general, we split $\mathbf{B}$ into blocks of $32 \times 32$ (this matches the fact that modern NVIDIA GPUs have 1024 threads per block). Then, we begin by fully computing all diagonal blocks in parallel. Notice that, due to the lower triangular structure of $\mathbf{B}$, those are fully independent. Similarly, each block depends exclusively on the blocks above it (up to the diagonal). Next, we perform an iterative process which will update all remaining blocks with respect to the new values of the most recently computed diagonal before completing the calculation of the blocks adjacent to it. This new diagonal is then the reference for the next iteration. Figure 3 visualizes this process. The calculation within the individual blocks is easy to parallelize. For more details, please refer to our code for this CUDA kernel. Overall, this way of calculating $\mathbf{B}$ is quite fast; e.g. for $m = 1024$, it takes $< 2$ milliseconds to execute on an NVIDIA RTX 3090 GPU.

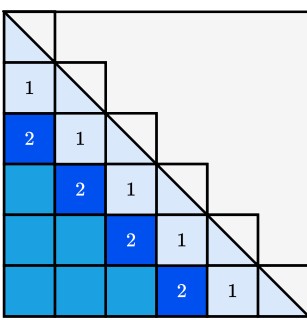

Figure 3: Visualization of the CUDA kernel for calculating $\mathbf{B}$. In each iteration, all blocks below the previous diagonal (here denoted as 1) are updated in parallel and those adjacent to the diagonal (here denoted as 2) are completed to form the reference diagonal for the next iteration.

**IHVPs**  After discussing an efficient implementation of the precomputation steps, we now focus on actually computing IHVPs. Algorithm 2 presents a vectorized implementation.

---

**Algorithm 2** Computing $\widehat{F}_m^{-1}\mathbf{x}$ in $O(dm + m^2)$ time, assuming that $\mathbf{D}$ and $\mathbf{B}$ have already been precomputed.

---

$\mathbf{q} \leftarrow \lambda^{-1}\mathbf{G}\mathbf{x}$
$q_1 \leftarrow q_1/(m + \mathbf{D}_{11})$
**for** $i \leftarrow 2, 3, \ldots, m$ **do**
    $\mathbf{q}_{i:} \leftarrow \mathbf{q}_{i:} - q_{i-1}\mathbf{D}_{i-1,i:}^\top$
**end for**
$\mathbf{q} \leftarrow \mathbf{q}/(m + \mathrm{diag}(\mathbf{D}))$
**return** $\lambda^{-1}\mathbf{x} - ((\mathbf{q}^\top\mathbf{B})\mathbf{G})^\top$

---

We note that if $\mathbf{x} = \nabla\ell_i$, then $\mathbf{q}_{:(i+1)} = \mathbf{D}_{:(i+1),i}/\mathrm{diag}(\mathbf{D}_{:(i+1)})$, i.e. the first $i$ elements in column $i$ of $\mathbf{D}$ divided element-wise by the corresponding diagonal values. This can be utilized for a more efficient combined update-multiply operation, which is essential for the M-FAC optimizer, where we first update our Hessian estimation with a gradient $\nabla\ell$ and then return the IVHP with this same

$\nabla\ell$. Further, reusing the same $\mathbf{p} = \mathbf{G}\nabla\ell$ for both updating $\mathbf{G}\mathbf{G}^\top$ as well as calculating $\mathbf{q}$, saves one redundant $O(dm)$ matrix-vector-product.

## C  Efficiently Implementing the Static Algorithm

Although the formulas discussed in the main body of the paper have low theoretical time complexity, a direct implementation does not make the best use of modern machine learning frameworks and GPUs. Therefore, we now present more efficient matrix-based versions. We note that a block-wise version would simply apply the techniques discussed here independently to each block.

Let $\mathbf{V} = [\mathbf{v_1}^\top; \mathbf{v_2}^\top; \ldots; \mathbf{v_m}^\top]$ be the $m \times d$ row-wise matrix of the $\mathbf{v}_i$ and $\mathbf{q} = (q_1, q_2, \ldots, q_m)^\top$ the $m$-vector of the $q_i$. Then, $\widehat{F}_m^{-1}\mathbf{x}$ can be written as follows, where $/$ denotes the element-wise division:

$$\widehat{F}_m^{-1}\mathbf{x} = \lambda^{-1}\mathbf{x} - \mathbf{V}^\top\left((\mathbf{V}\mathbf{x})/\mathbf{q}\right). \tag{24}$$

It is also possible to extract one element of each matrix row simultaneously (in terms of vectorized operations) with $O(dm)$ cost, for example the entire diagonal. Let $\boldsymbol{\pi}(\mathbf{x})$ be a mapping such that $\boldsymbol{\pi}(\mathbf{x})_i = x_j$ where $j$ is the column of the element to select in row $i$, e.g. $\boldsymbol{\pi}(\mathbf{x}) = \mathbf{x}$ to get the diagonal, and $\mathbf{e}^\pi$ a vector such that $e_i^\pi = 1$ if $\boldsymbol{\pi}(\mathbf{x})_i = x_i$ and $e_i^\pi = 0$ otherwise. Using these definitions, the calculation of the desired result $\mathbf{y}$ is described below, where $\odot$ denotes the element-wise product.

$$\mathbf{y} = \lambda^{-1}\mathbf{e}^\pi - \sum_{i=1}^{m} \frac{1}{q_k}\mathbf{v_i} \odot \boldsymbol{\pi}(\mathbf{v_i}) \tag{25}$$

Our implementation contains several additional optimizations. For instance, we can efficiently precompute $\mathbf{V}$ and $\mathbf{q}$ by repeatedly applying (24) to produce the next $\mathbf{v_i}$. In the next section, we additionally discuss several memory-saving optimizations, and in particular explicit page swapping between CPU and GPU memory for situations where $\mathbf{V}$ does not fully fit in the GPU memory.

Finally, Algorithm 3 demonstrates, using a Python-like matrix indexing syntax, how to efficiently precompute $\mathbf{V}$ and $\mathbf{q}$ by repeatedly applying (24) to produce the next $\mathbf{v_i}$. It should be noted that the row-wise matrix of gradients $\mathbf{G} = [\nabla\ell_1^\top; \nabla\ell_2^\top; \ldots; \nabla\ell_m^\top]$ is re-purposed as $\mathbf{V}$, a simple way to halve the peak memory consumption in practice.

---

**Algorithm 3** Precomputation of $\mathbf{V}$ and $\mathbf{q}$ given $\mathbf{G}$.

$\quad \mathbf{V} \leftarrow \mathbf{G}$
$\quad \mathbf{g} \leftarrow \mathbf{V}_{1,:}^\top$
$\quad \mathbf{V}_{1,:} \leftarrow \lambda^{-1}\mathbf{g}$
$\quad q_1 \leftarrow \mathbf{V}_{1,:}\,\mathbf{g}$
$\quad \mathbf{for}\ i \leftarrow 2, 3 \ldots, m\ \mathbf{do}$
$\quad\quad \mathbf{g} \leftarrow \mathbf{V}_{i,:}^\top$
$\quad\quad \mathbf{V}_{i,:} \leftarrow \lambda^{-1}\mathbf{g}^\top - ((\mathbf{V}_{:i,:}\,\mathbf{g})/\mathbf{q})^\top \mathbf{V}_{:i,:}$
$\quad\quad q_i \leftarrow m + \mathbf{V}_{i,:}\,\mathbf{g}$
$\quad \mathbf{end\ for}$

---

### C.1  Additional Optimizations

As we have seen, obtaining a good Fisher approximation of the Hessian requires a sizable number of gradients, which means that, for bigger networks, it can easily happen that the collection of all gradients used for Fisher estimation does not fully fit into GPU memory. We now discuss how to efficiently handle such situations, with an implementation that performs explicit swapping of gradient blocks between GPU and CPU memory (RAM). The most important steps of the method to be discussed are also visualized in Figure 4.

In general, a simple trick to halve the peak memory consumption of the static algorithm is repurposing the gradient matrix $\mathbf{G}$ as $\mathbf{V}$, which is possible as $\nabla\ell_i$ is not needed anymore after $\mathbf{v}_i$ and $q_i$ have been calculated. Now, for the explicit swapping implementation, we first split the collection of $m$ gradients into $k$ blocks / pages $\mathbf{G}^i$ containing at most $m/k$ gradients each. Those are then turned into the corresponding $\mathbf{V}^i$ blocks in increasing order of $i$. Additionally, we maintain a single buffer block $\mathbf{B}$ of the same size for accumulating intermediate results. All blocks reside in CPU memory and are only loaded to the GPU as needed, meanwhile $\mathbf{q}$ is small enough to be kept in GPU memory at all times. To compute $\mathbf{V}^i$ we first load block $\mathbf{V}^1$ fully into GPU memory. Then we load the first

gradient of $\mathbf{G}^i$ denoted by $\mathbf{g}_1^i$ and compute the corresponding $\mathbf{v}_1^1$ with respect to the loaded $\mathbf{V}^1$, which is afterwards saved in the first index of the buffer $\mathbf{b}_1$. After repeating this for all $\mathbf{g}_j^i$, block $\mathbf{V}^1$ is swapped with block $\mathbf{V}^2$ and the whole process starts again, accumulating the resulting $\mathbf{v}_j^2$ into the buffer. Eventually, after handling $\mathbf{V}^{i-1}$, we can load $\mathbf{B}$ into GPU memory, finish the calculation of the $\mathbf{v}_j^i$ and store them in $\mathbf{V}^i$ (reusing the memory of $\mathbf{G}^i$). It should be noted that the loading of $\mathbf{g}_j^i$ can be parallelized with the calculation of $\mathbf{g}_{j-1}^i$ and thus costs almost no extra time. Finally, one wants to choose the number of blocks $k$ to be as small as possible, to minimize the overhead caused by the $O(k^2)$ page swaps.

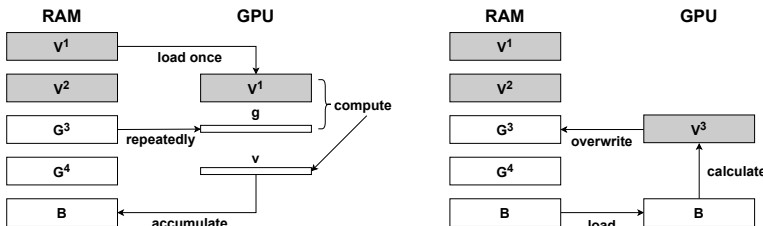

Figure 4: Visualization of the swapping implementation of the static algorithm, grey blocks have already been fully calculated. **Left**: $g$ is loaded from block $\mathbf{G}^3$, which allows calculating $\mathbf{v}$ using the GPU block $\mathbf{V}^1$ and then accumulating it into the buffer $\mathbf{B}$. **Right**: Calculation of $\mathbf{V}^3$ is finished using the completed buffer.

Overall, the implementation described above is effective in practice and allows scaling up our static algorithm (with only modest overhead over a pure GPU implementation), in its full non-blockwise version, to a large number of gradients, even for relatively large networks such as ResNet50.

Lastly, we note that, for a block-wise implementation, where a single block and all its corresponding gradient parts, easily fit into GPU memory (i.e. the opposite of the situation described thus far in this section), it can be beneficial to handle multiple blocks simultaneously via batch-matrix-multiplies. We provide an implementation that can do this, leading to very significant speed-ups on lower block-sizes where otherwise most of the time would be spent constantly loading in new memory from the CPU.

## D Additional Experimental Results

### D.1 Pruning Update Approximation

As discussed in the main paper, we use the OBS pruning framework but prune multiple weights at the same time, which as pointed out by [48], in extreme cases, is not guaranteed to be better than OBD due to not taking into account potential correlations. Thus, we now explore how much of a problem this is in practice. To do that, we compare three different M-FAC variations: not updating the non-pruned weights at all (i.e. OBD, but with a better Hessian approximation than the diagonal), jointly applying the OBS update for all pruned weights (i.e. the OBS approximation) and applying the correct simultaneous update by solving a large linear system (see [41] for details). We prune a ResNet20 / CIFAR-10 model in steps of 10% down to 90% sparsity, using three variations of the full block size M-FAC with $m = 1024$ while recomputing the inverse Hessian approximation after every pruning step. The recomputations ensure that we have a reasonably accurate loss approximation at every pruning step and that the number of weights pruned at each step is small enough to solve the corresponding linear system. Table 6 provides the results.

| Method | 10% | 20% | 30% | 40% | 50% | 60% | 70% | 80% | 90% |
|---|---|---|---|---|---|---|---|---|---|
| No update (OBD) | 91.4 | 91.3 | 91.0 | 90.2 | 88.4 | 84.8 | 77.2 | 50.4 | 11.5 |
| Simultaneous OBS | 91.4 | 91.4 | 91.5 | 91.2 | 90.8 | 89.7 | 86.8 | 75.6 | 28.2 |
| Linear solving | 91.5 | 91.4 | 91.4 | 91.2 | 90.7 | 89.7 | 87.5 | 75.7 | 28.1 |

Table 6: Comparing ResNet20 / CIFAR-10 accuracies of different M-FAC variations at varying levels of sparsity.

The results show a very clear gap in accuracy between the no-update and the approximate update version. At the same time, there appears to be almost no difference between the simultaneous OBS and the true OBS update, which has to solve a (potentially very large) linear system. This suggests

that the simultaneous update approximation done for computational tractability is also a reasonable choice in terms of achieved accuracy.

## D.2 Ablation Studies for One-Shot Pruning

To further examine the properties of the local loss approximation, we now present ablation studies on pretrained ResNet50/ImageNet and MobileNetV1-STR/ImageNet models. Experiments perform one-shot pruning according to the OBS metric estimated using M-FAC, with varying block size and number of gradients. The dampening factor $\lambda$ in these experiments is set to $10^{-5}$. Following [41], we used batched gradients (of size 16) as single samples inside the Fisher approximation. (This does not alter results, but reduces variance.)

The goal of these experiments is to examine two questions: 1) does larger block size always imply a better approximation and 2) does higher number of gradients always imply a better approximation? We will see that neither of these questions have obvious answers.

*The numbers presented for our one-shot experiments are the averages over two runs. As the variance is very small, we omit error bars.* We sometimes break the standard convention and "zoom into" the y axis for visibility.

### D.2.1 ResNet50 / ImageNet

The first set of experiments examines the dependency on the number of gradients and block size for the ResNet50 / ImageNet model. The left subfigure in Figure 5 shows results for block sizes between $2K$ and all weights, i.e. $d$, for a fixed number of $1K$ gradients, while the right subfigure shows the same, but for $2K$ gradients. The first graph presents a fairly unintuitive finding, i.e. that *lower* block sizes appear to provide better pruning accuracy. We analyze this finding in detail and explain it in Section D.3: roughly, this is due to the fact that gradient entries are scaled by the gradient norm over the block. As predicted by our analysis, this effect is mitigated by increasing the number of gradients used for estimation, in which case block size $10K$ yields the best results, and the performance of full block size also improves. Please see Section D.3 for the full analysis.

Figure 6 examines the complementary effect, that of number of gradients for a fixed block size (10K and 100K, respectively). The results suggest that more gradients help improve the estimation of the Fisher matrix, although we observe a saturation effect. We also note that, at higher one-shot sparsities (e.g., 60% in one-shot, not shown for visibility), this effect does not always hold. However, for such large pruning "perturbations" the pruning assumption that the Hessian is constant along the direction of pruning is unlikely to hold, which affects the stability of the results.

### D.2.2 MobileNetV1-STR / ImageNet

The second set of experiments shows the dependency on the number of gradients and block size for the MobileNetV1-STR / ImageNet model. We use the implementation and pre-trained weights from [24]. Figure 7 shows results for block sizes between $2K$ and all weights, i.e. $d$, for a fixed number of gradients ($1K$ in the left subfigure and $2K$ in the right subfigure). Again, in both cases, for this model *lower* block sizes appear to provide an improved approximation. This is most likely due to the gradient scaling effects which we discuss in Section D.3. However, the results show that for this compact model these effects are more prevalent than for the ResNet50 model.

Figure 8 examines the opposite effect, that of the number of gradients for a fixed block size ($10K$ in the left subfigure and $100K$ in the right subfigure). We show results for the number of gradients varying between $50$ and $4K$. The results clearly suggest that more gradients help improve the accuracy, although the improvement appears to saturate, e.g. between $2K$ and $4K$ gradients.

### D.2.3 Normalizer-Free Nets

We now examine the "compressibility" of the recently-proposed normalizer-free nets, which have shown competitive accuracy based on a similar structure to standard residual networks but without the batch normalization component [6]. We use the PyTorch re-implementation of [49]. In Figure 9a, we provide a relative comparison in terms of pruned accuracy between a normalizer-free and a regular version of ResNet50. These two networks have virtually the same number of parameters (approximately 25M); however, we notice that the normalizer-free variant is significantly "easier" to prune, in terms of the relative accuracy drop versus the dense baseline. We conjecture that this is because of the elimination of the *BatchNorm* layers. Specifically, when performing large one-shot pruning steps, the BatchNorm statistics become invalid following a pruning step, which can lead to

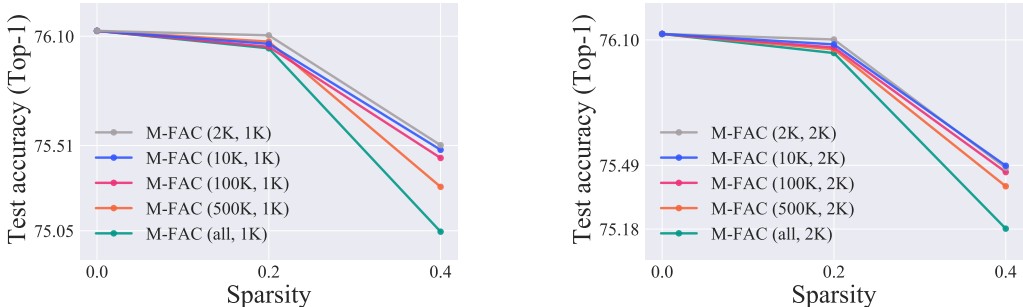

Figure 5: An ablation study on the ResNet50 / ImageNet model showing the effect of the block size for a fixed number of gradients.

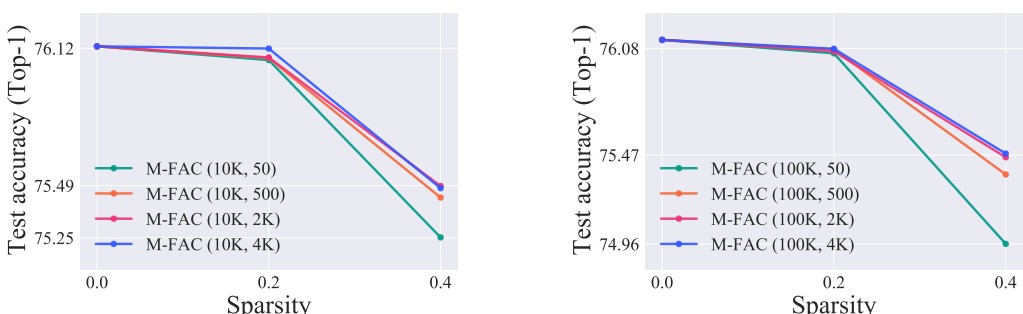

Figure 6: An ablation study on the ResNet50 / ImageNet model showing the effect of the number of gradients for a fixed block size.

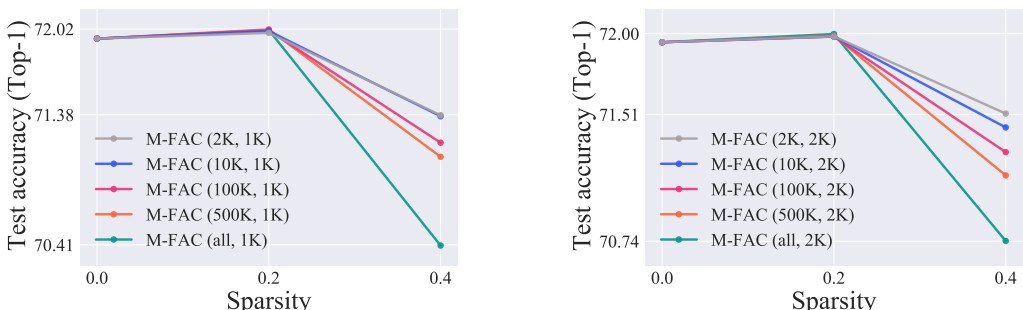

Figure 7: An ablation study on the MobileNetV1-STR / ImageNet model showing the effect of the block size for a fixed number of gradients.

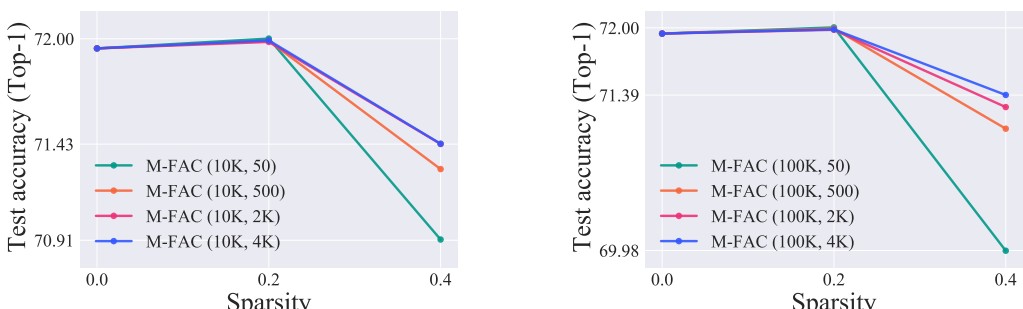

Figure 8: An ablation study on the MobileNetV1-STR / ImageNet model showing the effect of the number of gradients for a fixed block size.

a significant loss of accuracy; moreover, their removal may render the Fisher approximation more accurate, as the model loss is more stable in the local neighborhood.

### D.2.4 YOLOv3

Next, we study the effect of one-shot pruning on the YOLOv3-SPP model [39] for object detection on the COCO dataset [27]. We use the state-of-the-art implementation of [10]. We one-shot prune this model using global magnitude (the only available baseline) and M-FAC with block size 50K and 1K gradients. (This parameter value would be infeasible for WoodFisher due to storage costs.) The results in terms of the standard mAP@0.5 metric are provided in Figure 9b, and show that this model is quite stable under pruning, and that M-FAC provides more accurate pruning for this model as well, relative to magnitude pruning.

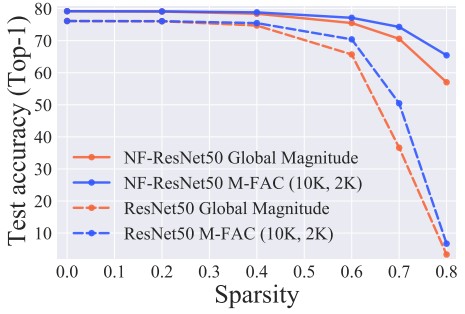

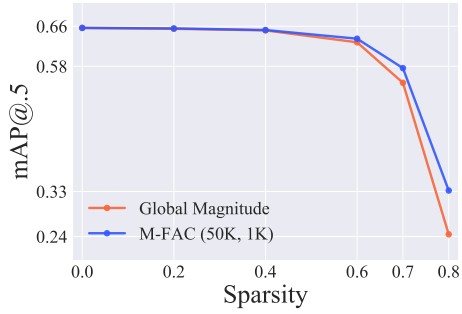

(a) Comparison between GMP and M-FAC pruning on the standard ResNet50 and a Normalizer-Free variant (NF-ResNet50). Observe the significantly better accuracy given by our method on the normalizer-free variant.

(b) Comparison between GMP and M-FAC in terms of the mAP@0.5 metric for a one-shot pruned complex YOLOv3-SPP / COCO model.

Figure 9: **Left**: The effectiveness of M-FAC pruning on Normalizer-Free ResNet50 versus the regular variant. **Right**: One-shot pruning comparison for the YOLOv3-SPP model on the large-scale object detection dataset COCO.

### D.2.5 K-FAC Pruning Comparison

In Figure 10a we compare pruning performance against a pruner which uses K-FAC to estimate second-order statistics (with and without dampening $\pi$) on a fully-connected network and the MNIST dataset. Notice the improved accuracy with M-FAC (and WoodFisher) methods compared to K-FAC, even in the setting with a small network on a simple task.

### D.2.6 Recomputation Effects

As noted before, our one-shot experiments stretch the OBD / OBS theory, as this approach implicitly assumes that the Hessian stays constant across the direction of pruning, which is unlikely to hold for very large pruning displacements. We can however examine the impact of this effect, by *recomputing* the Hessian along the direction of pruning. Figure 10b shows the effect of recomputation for the ResNet50 / ImageNet model, for 5 recomputation steps, uniformly across the pruning direction. Notice the significant increase in accuracy for the resulting sparse models.

### D.3 Normalization Effects

We now discuss our finding that, in the case of some models, e.g. MobileNetV1, smaller blocks appear to yield better accuracy. This can be explained by examining the recursive form of the elements of the diagonal inverse. Specifically, without blocking, we get that the $i$th diagonal element has the form

$$[\widehat{\mathbf{F}}^{-1}]_{ii} \simeq \frac{1}{\lambda} \left[ 1 - \frac{\nabla \ell_{1_i}^2}{\|\nabla \ell_1\|^2} - \frac{\left(\nabla \ell_{2_i} - \nabla \ell_{1_i} \frac{\nabla \ell_1^\top \nabla \ell_2}{\|\nabla \ell_1\|^2}\right)^2}{\|\nabla \ell_2\|^2 - \frac{(\nabla \ell_1^\top \nabla \ell_2)^2}{\|\nabla \ell_1\|^2}} - \cdots \right], \tag{26}$$

where $\nabla \ell_{1_i}$ and $\nabla \ell_{2_i}$ represent the $i$th element of a gradient. Specifically, notice that, in the case of "full" block size ("all" or $d$), the $i$th gradient entry is divided by the full gradient norm, which may cause it to become negligible if the norm is large. In turn, this leads to essentially the magnitude pruning ranking and update. By contrast, in the case of smaller blocks, the entry is only divided by the norm of the gradient over the block, which mitigates this normalization issue. Similarly, using

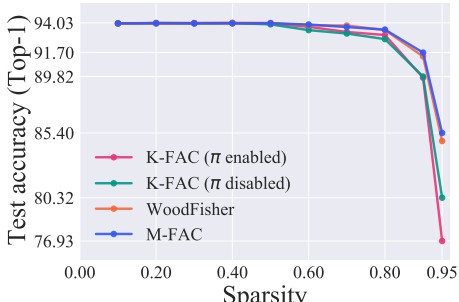
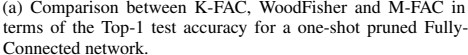
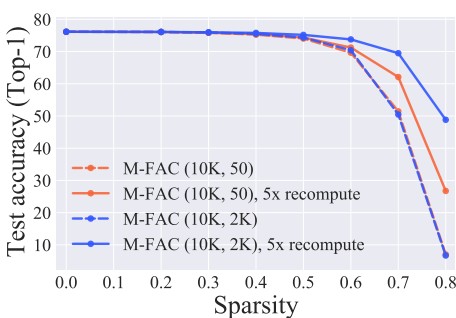

(a) Comparison between K-FAC, WoodFisher and M-FAC in terms of the Top-1 test accuracy for a one-shot pruned Fully-Connected network.

(b) The effect of a recomputation on the accuracy of pruned ResNet50/ImageNet model. Observe the significantly better accuracy provided by recomputation, especially at highwe number of gradients.

Figure 10: **Left**: One-shot pruning comparison for the fully-connected network on the MNIST dataset. **Right**: The effect of recomputation along the direction of pruning for ResNet50 / ImageNet.

more gradients also helps, since there are more summands in the above expression, which allows us to deviate from the magnitude pruning baseline.

### D.4 Pruning Experiments Hyperparameters

#### D.4.1 Gradual Pruning Comparisons

We begin by stating hyperparameters for our gradual pruning runs. For both MobileNetV1-STR and ResNet50-STR, we employ *identical* hyperparameter values to WoodFisher [41], so that our results are fully comparable. The only parameters we change are the ones which pertain to our algorithm, in particular *block size* and *number of gradients* $m$.

For the MobileNetV1-STR gradual experiments, pruning starts from the fully-trained model used by STR / WoodFisher, whose Top-1 validation accuracy is 72%. Gradual pruning is then performed over a total of 24 epochs, pruning additional weights every 4 epochs, followed by fine-tuning until the next pruning step. Pruning targets are set following the standard polynomial schedule of [55, 12]. Unless otherwise stated, SGD is used for fine-tuning. The learning rate during pruning is 0.005, momentum is set to 0.9, and weight decay is 0.0001. Following the last pruning step, fine-tuning is applied until epoch 100. The initial learning rate value of 0.005 is decayed multiplicatively by a factor of 0.92, every epoch, starting from the epoch 30.

For ResNet50-STR gradual experiments, pruning starts from the fully-trained model used by STR/WoodFisher, whose Top-1 validation accuracy is 77.01%. Gradual pruning is then performed over a total of 40 epochs, pruning additional weights every 5 epochs, followed by fine-tuning until the next pruning step. Pruning targets are set following the standard polynomial schedule of [55, 12]. Unless otherwise stated, SGD is used for fine-tuning. The learning rate during pruning is 0.005, momentum is set to 0.9, and weight decay is 0.0001. Following the last pruning step, fine-tuning is applied until epoch 100. The initial learning rate value of 0.005 is decayed multiplicatively by a factor of 0.6, every 6 epochs, starting from epoch 40 until epoch 90.

#### D.4.2 Practical Pruning

In our practical pruning experiments we use a different setup, which is optimized for the particular capabilities of M-FAC. We note that the high speed of M-FAC also makes more careful parameter tuning significantly easier.

In general, we always use M-FAC with block size 128 and 64 gradients (from a batch of 32 samples). Further, each pruning step is performed with 16 recomputations, which always prune the same fraction of remaining weights. This fraction can be calculated as $(i/t)^{1/k}$ where $i$ is the initial sparsity, $t$ is the target sparsity and $k$ is the number of recomputations. We immediately prune down all models to 50% sparsity before epoch 0 (we found that this sparsity is very easy to recover from, so intermediate steps are unnecessary). Further, we always use SGD without weight decay for a total of 100 epochs and we reset momentum after each pruning step.

For MobileNetV1 we perform a total of 15 additional pruning steps (not counting the initial pruning) which are performed with 3 epochs finetuning in between. All pruning steps prune the same fraction

of remaining weights (calculated by the formula described in the previous paragraph). The 3 epochs finetuning in between pruning steps all use the following learning rates $0.005, 0.005, 0.0005$, there is no additional decay (we noticed that dropping the learning rate in the last epoch usually results in a more accurate model as a starting point for the next pruning step). Finally, after the last pruning step at epoch 45, we finetuning for a total of 55 epochs with learning rate $0.005$ with drops by a factor of 10 at epochs 75 and 90.

For Resnet50 we perform a total of 20 additional pruning steps which 4 epochs finetuning at learning rates $0.005, 0.005, 0.005, 0.0005$ in between (due to the higher target sparsities, individual steps are bigger and the model seemed to need more time to recover well in between steps). We first calculate a 15-step equal-fraction schedule to the final target sparsity, execute the first 10 steps and then replace the last 5 with a 10-step equal-fraction schedule from the current to the target sparsity (due to the very high target sparsities the last steps needed to be smaller). Eventually, we finetune for 20 epochs with learning rate .005 dropped by a factor of 10 at epochs 90 and 95.

### D.5 Optimization Experiment Hyperparameters

**ResNet20/32**   We now discuss the hyperparameters used for the ResNet20/32 comparison between M-FAC and various first and second order optimizers. Both models are trained with batch-size 128 for 164 epochs ($\approx 64000$ steps) while dropping the learning rate by a factor of 0.1 after $50\%$ and $75\%$ of training. This is exactly the training setup used by [17]. For M-FAC, we use $m = 512$ gradients, dampening $\lambda = 10^{-5}$ and initial learning rate $10^{-3}$, which were determined to be reasonable default values during development. For the other methods we use tuned initial learning rates from literature; in particular the value for SGD is from [17] while the values for Adam, AdamW and AdaHessian are from [52]. Since we found that the exact weight decay value can have a significant impact on the final test accuracies, we performed grid-searches over the commonly used values $\{0, 10^{-5}, 10^{-4}, 5 \cdot 10^{-4}, 10^{-3}, 3 \cdot 10^{-3}, 10^{-2}\}$ for all methods that use weight decay. Table 7 summarizes the final hyper-parameter settings.

| Method | learning rate | momentum | weight decay |
|--------|---------------|----------|--------------|
| SGD | 0.1 | 0.9 | 0.0001 |
| Adam | 0.001 | (0.9, 0.999) | – |
| AdamW | 0.01 | (0.9, 0.999) | 0.01 |
| AdaHessian | 0.015 | (0.9, 0.999) | 0.003 (RN20), 0.0005 (RN32) |
| M-FAC ($m = 512$) | 0.001 | – | 0.003 |

Table 7: Hyperparameter settings used for the ResNet20 (RN20) and ResNet32 (RN32) experiments.

All experiments were repeated 5 times with different random seeds and we report the median of the best test accuracy; the standard deviations were generally quite low at around 0.2 percent accuracy.

**Hyperparameters for GGT and K-FAC.**   Unfortunately, GGT and K-FAC did not produce reasonable results in the setup discussed so far (i.e. achieved poor accuracy or diverged). Thus, for fairness, we decided to adopt the recommendations of the method authors, even if those were no longer exactly comparable with the other experiments (e.g. different batch-size, learning rate schedule, etc.). Further, we also performed considerably more extensive hyper-parameter searches for these methods.

For K-FAC, we use the authors' carefully-tuned parameters for ResNet20, which they published in their official repository[1]. For ResNet32, we adopt the same parameters, but use an initial learning rate of 0.075 and an initial dampening of $\approx 0.1887$, which we identified to work best for this model via grid search (see Table8 for the grid). The biggest differences of this setup compared to M-FAC are the batch-size of 1000 and the smoothly exponential decaying learning rate.

For GGT, we use a batch-size of 128 and a gradient window size of $r = 100$ (which is $> 5\times$ lower than for M-FAC, however we surprisingly found that GGT's performance, unlike M-FAC's, did not improve with larger $r$ and thus kept this as the best value). Further, we use an initial learning rate of 0.05 and a cosine decaying schedule with $T = 40$, which we found via grid search. All GGT experiments were performed with the author's implementation contributed to TensorFlow. Our GGT search grid is shown in Table 9.

---

[1] https://github.com/tensorflow/kfac

| Init LR | $0.5, 0.25, 0.1, 0.05, 0.025, 0.01, 0.005, 0.0025, 0.001$ |
|---|---|
| Final LR | $10^{-4}, 10^{-5}$ |
| Init Dampening | $0.3887, 0.2887, 0.1887$ |
| Inversion freq. | $1, 10$ |

Table 8: ResNet32 K-FAC search grid; parameters not listed here were kept the same as the ResNet20 settings.

| Init LR | $0.5, 0.1, 0.05, 0.01, 0.005, 0.001, 0.0005, 0.0001$ |
|---|---|
| Window size $r$ | $50, 100, 200, 500$ |

Table 9: GGT search grid; parameters not listed here were kept at the author's recommended settings.

**Wide ResNet / MobileNetV1.**   Next, we list the exact hyperparameters used for optimizing Wide ResNet (WRN) and MobileNetV1 models. The corresponding experiments were designed to explore how well M-FAC performs relative to other methods without any parameter tuning, i.e. just using reasonable default values. In the case of Wide ResNet, all models are trained for 200 epochs with batch size 256, where the learning rate is dropped by a factor 10 after 50% and 75% of training. SGD uses an initial learning rate of 0.1 and momentum of 0.9 (as suggested by [53]). Adam runs with default parameters, i.e. a learning rate of 0.001, $\beta_1 = 0.9$ and $\beta_2 = 0.999$. M-FAC uses $m = 1024$ and the standard learning rate of 0.001 (but no momentum). No method uses any weight decay in these experiments. For MobileNetV1, we use exactly the same optimizer settings and learning rate schedule, but we train only for 100 epochs.

**Sparse Finetuning.**   For the sparse fine-tuning experiments, pruning occurs identically for all methods. However, we fine-tune the models using either SGD or M-FAC with $m = 512$, for 30 epochs. Both fine-tuning algorithms are run with exactly the same hyperparameter values: initial learning rate 0.005, reduced every 10 epochs by a factor of 10 and batch-size 256.

## D.6   Wall-Clock Time Comparison.

In Figures 11 and 12 we show the test accuracies during the training of Wide ResNet (WRN) on the CIFAR-100 dataset and MobileNetV1 on the ImageNet dataset with SGD, Adam and M-FAC with respect to the wall clock time, when executed on a single NVIDIA RTX 3090 GPU. (In the case of the largest network, WRN 22-4, we use a second GPU's memory to store gradients, but still use only a single GPU for computation.) One can see that, in most plots, M-FAC reaches better accuracies than SGD already after only slightly higher training time. Further, we can see that, for the WRN model, the early training accuracy increases fastest for M-FAC even with respect to the wall-clock time, a phenomenon often observed with methods which try to leverage approximate second-order information.

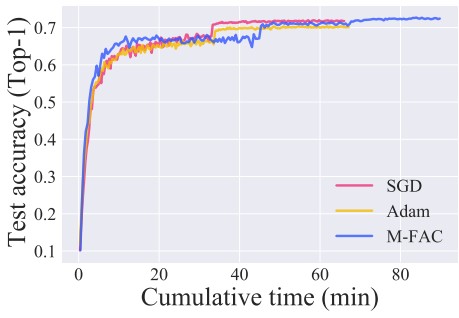
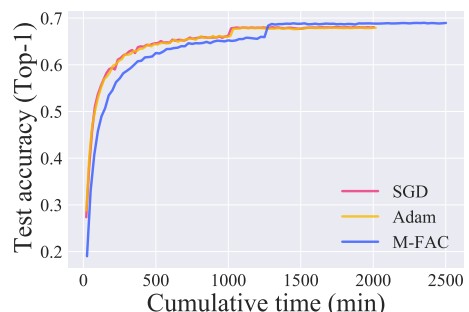

(a) Accuracy versus time for WRN 40-2 / CIFAR-100.  (b) Accuracy versus time for MobileNetV1 / ImageNet.

Figure 11: Optimization results in accuracy-versus-time format for WRN40-2 and MobileNetV1.

## D.7   Cosine Similarity of Descent Directions

Finally, in Figure 13, we examine the quality of the sliding window approximation to the Fisher matrix. We run optimization with M-FAC on ResNet20 / CIFAR-10 (using $m = 512$ and otherwise

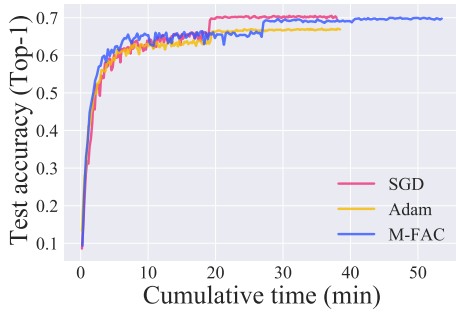

(a) Accuracy versus time for WRN 22-2 / CIFAR-100.

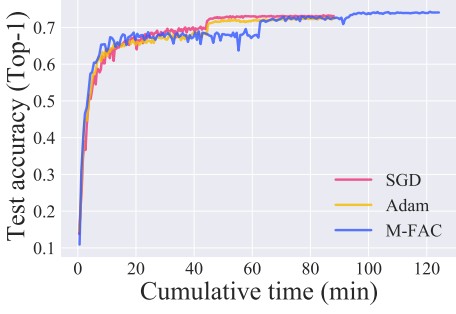

(b) Accuracy versus time for WRN 22-4 / CIFAR-100.

Figure 12: Optimization results in accuracy-versus-time format for Wide Residual Networks (WRN).

the same hyperparameters as discussed in the previous section), and every 512 steps we sample $2 \times 512$ gradients to produce 2 static estimates of the Fisher matrix, and then we compare the cosine similarity of the descent direction given by the dynamic algorithm at this step, $\widehat{F}^{-1}_{dynamic} \cdot \nabla \ell$ with $\widehat{F}^{-1}_{static1} \cdot \nabla \ell$ (denoted as dynamic–static) as well as the cosine similarity between $\widehat{F}^{-1}_{static1} \cdot \nabla \ell$ and $\widehat{F}^{-1}_{static2} \cdot \nabla \ell$ (denoted as static–static).

The results show that: 1) the cosine similarities between the sliding window approximation and the "fresh" approximation are extremely close; 2) they tend to improve significantly as we advance in the optimization process. Overall, this validates the sliding window approximation made by our method.

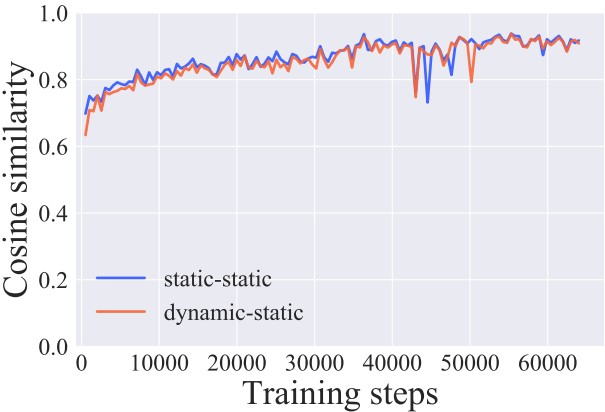

Figure 13: Cosine similarities over the course of training a ResNet20 / CIFAR-10 model. The directions are remarkably close for the dynamic algorithm, suggesting that the sliding window approximation is valid.

# E   Transformer Natural Language Modelling

In this section, we provide more detailed results of our natural language modelling experiments for the tiny (t) and mini (m) variants of the BERT Transformer model. As already stated in the main paper, M-FAC always uses $m = 1024$ gradients (except if a task has significantly less total training batches), dampening $\lambda = 10^{-6}$, learning rate $10^{-4}$ and no weight decay or momentum. We preserve all other configurations (e.g. batchsize, number of epochs, learning rate schedule) from the baseline. The SQuADv2 HuggingFace Adam baseline[2] uses initial learning rate $3 \cdot 10^{-5}$ and trains for 2 epochs, while for the GLUE tasks, it[3] uses initial learning rate $2 \cdot 10^{-5}$ and trains for 3 epochs (5 epochs for MRPC). Table 10 and Table 11 show our detailed (including all task metrics and standard deviations)

---

[2] https://github.com/huggingface/transformers/tree/master/examples/pytorch/question-answering, accessed: 2021-10-26

[3] https://github.com/huggingface/transformers/tree/master/examples/pytorch/text-classification, accessed: 2021-10-26

question answering SQuADv2 and text-classification GLUE results, respectively. At last, in Table 12, we compare BERT-tiny models optimized with M-FAC against tuned AdamW results (both trained for 4 epochs) by the BERT authors[4], on the GLUE test sets. We find that M-FAC (sometimes with modest tuning) can also outperform this competitive baseline on most tasks, even without using any weight decay or momentum.

|   |       | SQuADv2 (EM) | SQuADv2 (F1) |
|---|-------|--------------|--------------|
| t | Adam  | $48.41 \pm 0.57$ | $49.99 \pm 0.54$ |
| t | M-FAC | $\mathbf{49.80 \pm 0.43}$ | $\mathbf{52.18 \pm 0.20}$ |
| m | Adam  | $54.80 \pm 0.47$ | $58.13 \pm 0.31$ |
| m | M-FAC | $\mathbf{58.02 \pm 0.39}$ | $\mathbf{61.35 \pm 0.24}$ |

Table 10: Comparing the M-FAC optimizer against HuggingFace's Adam baselines on the SQuADv2 dataset.

|           | SST-2 (Ac) | MRPC (F1) | MRPC (Ac) | STS-B (Pe) | STS-B (Sp) |
|-----------|------------|-----------|-----------|------------|------------|
| t / Adam  | $80.11 \pm 0.65$ | $81.68 \pm 0.33$ | $69.90 \pm 0.32$ | $64.39 \pm 5.02$ | $66.52 \pm 5.67$ |
| t / M-FAC | $\mathbf{81.86 \pm 0.76}$ | $\mathbf{82.77 \pm 0.22}$ | $\mathbf{72.94 \pm 0.37}$ | $\mathbf{80.15 \pm 0.52}$ | $\mathbf{80.62 \pm 0.43}$ |
| m / Adam  | $\mathbf{85.46 \pm 0.58}$ | $84.57 \pm 0.36$ | $76.57 \pm 0.80$ | $82.09 \pm 0.54$ | $82.64 \pm 0.71$ |
| m / M-FAC | $84.20 \pm 0.58$ | $\mathbf{85.06 \pm 1.63}$ | $\mathbf{78.87 \pm 2.33}$ | $\mathbf{84.66 \pm 0.30}$ | $\mathbf{84.65 \pm 0.30}$ |

|           | QQP (F1) | QQP (Ac) | MNLI-m (Ac) | MNLI-mm (Ac) | QNLI (Ac) |
|-----------|----------|----------|-------------|--------------|-----------|
| t / Adam  | $77.58 \pm 0.08$ | $81.09 \pm 0.15$ | $65.36 \pm 0.13$ | $66.78 \pm 0.15$ | $77.85 \pm 0.15$ |
| t / M-FAC | $\mathbf{79.71 \pm 0.13}$ | $\mathbf{84.29 \pm 0.08}$ | $\mathbf{68.28 \pm 3.29}$ | $\mathbf{68.98 \pm 3.05}$ | $\mathbf{81.17 \pm 0.43}$ |
| m / Adam  | $82.43 \pm 0.10$ | $86.45 \pm 0.12$ | $73.30 \pm 0.20$ | $74.85 \pm 0.09$ | $\mathbf{83.85 \pm 0.10}$ |
| m / M-FAC | $\mathbf{82.67 \pm 0.23}$ | $\mathbf{86.75 \pm 0.20}$ | $\mathbf{74.59 \pm 0.41}$ | $\mathbf{75.95 \pm 0.14}$ | $83.70 \pm 0.13$ |

Table 11: Comparing the M-FAC optimizer against HuggingFace's Adam baselines on the GLUE benchmark suite (we show scores on the validation set).

|        | SST-2 (Ac) | MRPC (F1) | MRPC (Ac) | STS-B (Pe) | STS-B (Sp) |
|--------|------------|-----------|-----------|------------|------------|
| AdamW  | 83.2 | 81.1 | 71.1 | 74.3 | **73.6** |
| M-FAC  | $\mathbf{83.4}^*$ | $\mathbf{81.9}^*$ | $\mathbf{72.7}^*$ | $\mathbf{75.3}^*$ | $73.2^*$ |

|        | QQP (F1) | QQP (Ac) | MNLI-m (Ac) | MNLI-mm (Ac) | QNLI (Ac) |
|--------|----------|----------|-------------|--------------|-----------|
| AdamW  | 62.2 | 83.4 | 70.2 | 70.3 | 81.5 |
| M-FAC  | **62.8** | **83.9** | **71.0** | **70.5** | **81.7** |

Table 12: Comparing M-FAC optimizer (without weight decay) against BERT authors' *tuned* BERT-tiny AdamW baseline on a subset of GLUE benchmark *test* sets. * Modest tuning of learning rate and dampening because of the very low number of samples / gradients in the training data.

---

[4] `https://github.com/google-research/bert`, accessed: 2021-10-26