# OpenReview forum: "M-FAC: Efficient Matrix-Free Approximations of Second-Order Information"
_NeurIPS.cc/2021/Conference — NeurIPS 2021 Poster_

### Official Review · Reviewer_zd2D · 2021-07-15

**Rating:** 6
**Confidence:** 5

**Summary:**

This paper concentrates on the fast implementation of computing IHVPs (Inverse-Hessian Vector Products, and the expression here means "Inverse-Empirical-Fisher Vector Products") in training DNN, whose high time and memory costs remain a core problem. This paper is mainly based on the WoodFisher method. The WoodFisher uses the SMW (Sherman-Morrison-Woodbury) formula to compute the inverse of Fisher. This paper proposes a new algorithm (M-FAC) to accelerate this process. With two algorithms (static and dynamic, which correspond to gradients from single and multiple epochs), M-FAC can avoid matrices operations and decrease the time and storage costs, and the final time complexity is linear-in-dimension. In real practice, M-FAC outperforms K-FAC and has a similar performance with AdaHessian. Besides, when pruning DNNs, M-FAC also shows an improvement compared with WoodFisher.

**Ethical Concerns:**

No ethical problem has been found

**Limitations And Societal Impact:**

No obvious societal impact can be predicted

**Main Review:**

This paper starts the discussion from the rank-one matrix updating rule of the  empirical Fisher matrix's inverse, which is deduced directly from the SMW formula. This updating rule is discussed by the WoodFisher method. This SMW trick may be intractable due to the prohibitive time cost O(d^2*m) (d denotes the dimension of the parameter space and m denotes the number of gradients), and M-FAC can alleviate this cost by storing the intermediate information when updating the inverse vector product. Moreover, M-FAC can also be applied to any situation where the rank-one matrix updating rule exists.

Specifically, M-FAC includes the static algorithm and the dynamic algorithm. The static algorithm is for the common mini-batch situation within a single epoch, and the dynamic algorithm is used when the curvature information is updated by the new gradient. The result is that the time complexity can be decreased to linear-in-dimension. These algorithms are correct, and the derivation process is clear and natural. One problem may be that the algorithm would still be time-consuming when the dimension(d) is very large, and M-FAC is based on the recursion and may accomplish by for loops, which can be slow in real practice without distribute computing. Another possible problem of the dynamic algorithm may be that it is limited to the process of updating only one gradient. This might lead to the lack of flexibility when tuning, as other second-order methods can decrease the frequency of updating the curvature information.

In the experiments part, M-FAC performs better than WoodFisher when pruning DNNs. However, this may be a little confusing, because M-FAC and WoodFisher both use the SMW trick and the computation is exact. In Appendix D.8, "The only parameters we change are the ones which pertain to our algorithm, in particular block size and number of gradients m", so maybe the difference is from this change. If the settings of two methods are the same, would M-FAC still outperform WoodFisher? If so, what is the reason behind? Besides, M-FAC outperforms K-FAC and has a similar performance with AdaHessian. Nevertheless, K-FAC can be better than SGD in some cases, as shown in "A Kronecker-factored approximate Fisher matrix for convolution layers (2016) " (http://proceedings.mlr.press/v48/grosse16.pdf). As the M-FAC paper shows the results with "Epoch" instead of "wall clock time", the result that K-FAC is obviously slower than SGD is confusing. Another problem is that the paper also conduct experiments with GGT, and it performs badly. This may not be consistent with the results shows in "Efficient Full-Matrix Adaptive Regularization (2019)"(http://proceedings.mlr.press/v97/agarwal19b/agarwal19b.pdf). As the performance of M-FAC is similar to SGD in Figure 1(b), this numerical result is not show strong improvement. Furthermore, while some experiments are accomplished on Pytorch and others are accomplished on TensorFlow, the comparison might not be persuasive. Additionally, the experiments are accomplished on the ResNet20/Cifar-10, and the author need to add more experiments on other large datasets.

It is recommended to add more related works, because the work on the updating rank-one matrix rule may be studied before. Some related works are also worth attention, like another approach to compute IHVPs in "Sketchy Empirical Natural Gradient Methods for Deep Learning(2020)"(https://arxiv.org/pdf/2006.05924.pdf). The algorithm containing the rank-one updating rule have been studied in "Scalable Adaptive Stochastic Optimization Using Random Projections (2016)"(https://arxiv.org/pdf/1611.06652.pdf).

Overall, this paper is well-written. Here are a few detailed suggestions:
* (Line 249) Should the first "\hat{F_i}^{-1}" be "\hat{F_i}" ?
* (Section 3) More explanations to the different application scenarios of the static and dynamic algorithms would help readers understand their differences.
* (Section 3.2) When showing the complexity analysis, it may be better to put the results in a table and add some comparisons with other second-order methods.
* (Section 4.2) As the M-FAC changes the computing rule in one epoch, the improvement of the time cost within one epoch compared with other methods may be more persuasive. The result with "wall clock time" can also help.
* (Table 4 Right) Add experiments with another second-order method (like K-FAC) would help readers understand the real improvement of the M-FAC.

After reading the detailed rebuttal, I am still concerned on the efficiency and the difference to WoodFisher.


**Time Spent Reviewing:**

12 hours

---

> ### Author Response · Authors · 2021-08-10
> **Response to Reviewer 4 zd2D**
>
> Thank you for your detailed suggestions and for checking the derivations!
>
> > One problem may be that the algorithm would still be time-consuming when the dimension(d) is very large, and M-FAC is based on the recursion and may accomplish by for loops, which can be slow in real practice without distribute computing.
>
> We are not sure we understand this point: 1) our algorithms execute sequential loops; and 2) all of our experiments execute the computation on a single GPU (in particular, the timing results in Table 2), so our results do not benefit from distributing computation (though, M-FAC in general could of course benefit). Finally, we note that the dimension $d$ enters the computations primarily in form of matrix-vector products, which are extremely efficient to execute.
>
> We would be happy to further engage on this point (or any others) during the discussion period.
>
> > Another possible problem of the dynamic algorithm may be that it is limited to the process of updating only one gradient. This might lead to the lack of flexibility when tuning, as other second-order methods can decrease the frequency of updating the curvature information.
>
> This is not the case: Notice that we can in fact replace $k$ gradients simultaneously in $O(dmk + m^3)$ time by updating the corresponding $k$ rows and columns of $GG^\top$ and then recomputing $D$ and $B$ _once_ (note that for small $k$, this will actually not be much slower than replacing a single gradient due to caching). With this, we can also decrease the update frequency of the M-FAC optimizer, replacing $k$ gradients every $k$ steps, and thereby improve overheads (e.g. to 1.15x for both RN20 and RN32 with $k = 8$) with only a small impact on the overall optimization performance (for example, we observed only ~0.25 lower accuracy in Top-1 on RN20).
> Thank you for bringing up this point, it is definitively a good idea to mention this in the next revision of the paper.
>
> > In the experiments part, M-FAC performs better than WoodFisher when pruning DNNs. However, this may be a little confusing, because M-FAC and WoodFisher both use the SMW trick and the computation is exact. In Appendix D.8, "The only parameters we change are the ones which pertain to our algorithm, in particular block size and number of gradients m", so maybe the difference is from this change. If the settings of two methods are the same, would M-FAC still outperform WoodFisher? If so, what is the reason behind?
>
> We believe there is an unfortunate confusion here. M-FAC and WoodFisher are indeed numerically equivalent at the same block size and number of gradients, so M-FAC would be identical to WF for the same values (barring sampling randomness). This is the reason we don’t show the two methods at the same parameter values, as the results perfectly overlap. The main advantage of M-FAC is that it is able to run parameter values which are either infeasible or too slow for WF (please see Table 2 for a running time comparison for various parameters). In turn, the improved approximation leads to non-trivial accuracy improvements for pruned models.
>
> > M-FAC outperforms K-FAC and has a similar performance with AdaHessian.
>
> It is true that the _accuracies_ of M-FAC and AdaHessian are similar, but we stress that AdaHessian is 2-3x _slower_ in terms of wall-clock time than M-FAC at similar accuracies. This is an inherent limitation of AdaHessian, as it has to perform double back-propagation and that thus its minimal overhead vs SGD is 2x, and is closer to 3x in practice.
>
> > Nevertheless, K-FAC can be better than SGD in some cases, as shown in "A Kronecker-factored approximate Fisher matrix for convolution layers (2016)"
>
> We fully agree that K-FAC can outperform SGD when carefully tuned. However, the results the reviewer is referencing are executed on a somewhat outdated experimental setup -- in particular, much better hyper-parameters are currently known for SGD. We compare against the best currently known hyper-parameters for SGD (learning rates and weight-decay values). For K-FAC, we used parameters which have already been “heavily tuned” by the K-FAC authors, as the authors themselves state in (https://github.com/tensorflow/kfac/blob/master/kfac/examples/keras/KFAC_vs_Adam_on_CIFAR10.ipynb). We used exactly their implementation. These reach similar accuracies to SGD on ResNet20 / CIFAR-10 (see Table 3). For ResNet-32, we started from the same parameters as for ResNet20, and performed manual tuning of the K-FAC hyper-parameters via grid search, until we could find no further improvement. We believe this is fair, since the architectures are quite similar and the task is the same, and therefore one would expect good hyper-parameters to be close.
>
> > As the M-FAC paper shows the results with "Epoch" instead of "wall clock time", the result that K-FAC is obviously slower than SGD is confusing.
>
> We are confused by the reviewer’s statement, as our paper _does not state_ that K-FAC is generally slower than SGD, and we _do_ provide wall-clock time results in Appendix D.10 for four different applications.
> Specifically, we stress that the overheads in Table 4 (left) are given relative to SGD running on exactly the same setup (i.e. framework and GPU) as the respective optimizer (and not relative to M-FAC). The overheads in brackets are again relative to SGD in the same setup, but on the same parameter configuration (i.e. batch / gradient window size) as used by M-FAC.
>
> > Another problem is that the paper also conduct experiments with GGT, and it performs badly. This may not be consistent with the results shows in "Efficient Full-Matrix Adaptive Regularization (2019)"
>
> We spent significant effort trying to perform a fair comparison with K-FAC and GGT, and we believe that we are the first to compare to both full-matrix adaptive methods (GGT) and with “natural gradient” methods (K-FAC). Here is our procedure for running and tuning GGT:
>
> - We followed all parameters from the paper and used the authors’ original Tensorflow implementation. As the implementation employs several additional heuristics, we believe it is fairer to use the original code.
> - As the paper did not contain all hyper-parameters for the provided runs, we contacted the authors to obtain them. Unfortunately there is no public version of the code with end-to-end runs, but the authors provided us with guidelines for how to choose missing hyper-parameters.
> - We followed their guidelines, and filled in missing pieces via grid search. (Please see the answer to Reviewer 2 6ujH for details on the search.)
> - We also expected to see GGT perform better, as it is quite an elegant approach. However, the results we show are fairly close in terms of trends to what is shown in the paper (in particular, the performance of GGT is similar to that of Adam).
>
> > As the performance of M-FAC is similar to SGD in Figure 1(b), this numerical result is not show strong improvement.
>
> Please examine Table 4 (left): it shows the numerical validation accuracies corresponding to Figure 1 (d). Please note that there is a fairly significant gap with SGD accuracy in this experiment. A similar trend is visible in Table 4 (right), but under a different tuning approach. In Appendix D10 we also provide wall-clock vs. accuracy plots for the experiments in Table 4 Right. Those show M-FAC learning the fastest in the first few epochs and reaching the highest test accuracy fastest or with only very slightly longer training time.
>
> > Furthermore, while some experiments are accomplished on Pytorch and others are accomplished on TensorFlow, the comparison might not be persuasive.
>
> This is done so we can always faithfully compare against the authors’ original implementation for K-FAC and GGT, which are in TensorFlow. For K-FAC, we have also run against a Pytorch version of the optimizer and obtained very similar results in terms of wall-clock time (but preferred to use the “official” version). Using the original code is important because optimizers use different additional heuristics for better accuracy; however, the exact framework is less important for performance: most of the computation happens in numerical (CUDA) calls, which are implemented using the same libraries in both frameworks. Also, we emphasize that we always compare speed w.r.t SGD executed in the same framework.
>
> > Additionally, the experiments are accomplished on the ResNet20/Cifar-10, and the author need to add more experiments on other large datasets.
>
> We emphasize that our submission does have numerical results on CIFAR-100 and ImageNet (Table 4 Right). For the rebuttal, we have also added optimization results on ResNet18/ImageNet and BERT (various natural language tasks), and pruning results for YOLOv5/COCO. Please see our answer to Reviewer 2 6ujH for training results on BERT models.
>
> ResNet18/ImageNet optimization, Top-1 accuracy, standard hyperparameters from Torchvision:
> SGD: 69.75; M-FAC: 69.84
>
> Pruning YOLOv5S (https://github.com/ultralytics/yolov5) to 80% average sparsity: 52.98 mAP (GMP) vs 53.8 mAP (M-FAC) vs 55.4 mAP (dense); mAP values at threshold 0.5
>
> > [Related Works]
>
> Thank you for these related work suggestions, which we will discuss in detail in the next version of the paper. In brief, these works are indeed related, but are not very close technically. The second paper is a precursor of GGT, using additional sketching approximations. The first reference aims to perform efficient low-rank updates at the layer level, via additional approximations, in particular sketching. We will add a precise discussion of both to our related work section.
>
> Thank you also for your other detailed suggestions, which we will carefully consider for the next revision. On Section 4.2, note that the overhead we give is in per-epoch wall-clock time relative to SGD; we will also make this more clear.

---

### Official Review · Reviewer_JscG · 2021-07-15

**Rating:** 7
**Confidence:** 4

**Summary:**

The authors propose a compute-/memory- efficient implementation of inverse empirical Fisher vector products (IEFVPs) for deep neural networks, called M-FAC, based on the Woodbury-Sherman-Morrison inversion formula. They use IEFVPs as estimations of inverse Hessian vector products (IHVPs) used in model compression (e.g., Optimal Brain Damage) and second-order optimization. M-FAC is similar to the previous method, WoodFisher (Singh and Alistarh, 2020), which uses the WSM inversion formula to calculate inverse empirical Fisher. Unlike WoodFisher (calculates matrices), M-FAC directly calculates IEFVPs (vectors) and thus requires less compute/memory cost. This allows M-FAC to calculates IEFVPs with the full empirical Fisher and a larger number of samples ($m$) than WoodFisher.
- The static algorithm (with a fixed set of samples) of M-FAC is tested on the one-shot/gradual pruning with trained CNNs on ImageNet. It shows a better accuracy-sparsity trade-off than WoodFisher and other magnitude-based methods.
- The dynamic algorithm (with the fixed size sliding window of samples) of M-FAC is tested on training CNNs on CIFAR-10 and fine-tuning sparse CNNs on CIFAR-100 and ImageNet with preconditioned SGD. It shows a better computational overhead (vs. SGD) than existing second-order optimization methods in training and better accuracy than first-order methods in fine-tuning.

**Limitations And Societal Impact:**

The authors adequately addressed the limitations of the presented work.

**Main Review:**

This paper is well-organized. The literature, algorithms, novelty, costs/limitations, and experiments are clearly explained.
The proposed M-FAC is a clear improvement over the previous WoodFisher algorithm for pruning DNN models.  However, I feel that there are intrinsic limitations of M-FAC (and methods based on the WSM inversion formula in general) regarding scalability and practicality, as I will describe below. For this reason, I believe this work does not contain enough ideas to be appreciated by the community, especially for practitioners. For the improvements, I would suggest exploring empirical/theoretical justification/insight of using a small sample size $m$ in pruning and optimization.

---------------

Comments on pruning DNNs with the static algorithm
- M-FAC is a natural, reasonable improvement over the WoodFisher.
  - It allows the use of a larger sample size $m$ and the information of the full empirical Fisher, which is a clear advantage.
  - As shown in the experiments, M-FAC can achieve a better accuracy-sparsity trade-off.
- However, as discussed by the authors, the cost of storing the $m$ gradients is a problem that we cannot ignore in practice.
  - If a model has massive parameters (and this is why we want to prune them), we can only use a very small $m$.
  - The goal of the OBD/OBS framework is to measure the sensitivity of each parameter to the empirical loss (with training set).
  - But we need to limit the sample size $m$ (e.g., 400, 1k, 4k) much smaller than the training set size (e.g., 1.28M for ImageNet).

Comments on optimization with the dynamic algorithm
- No error bars (by varying initialization points) are reported, so it is unclear whether M-FAC shows better accuracy by chance.
- Regarding the evaluation of second-order methods
  - Second-order optimizers with rich (not diagonal) curvature information (e.g., K-FAC) tend to show faster convergence (in terms of epochs) than first-order optimizers.
  - But no such fast convergence is observed in the experiments.
  - I believe the benefit of second-order optimization is a faster convergence (we have theoretical justification/insight at least in convex problems) rather than better test accuracy (actually, we have the opposite insights, e.g., [1][2]).
  - I feel it is not appealing to claim that the proposed second-order method achieves better test accuracy without theoretical justification/insight because it doesn't ensure success in general settings.
- Due to the recursive nature of M-FAC (and the WSM inversion formula), it doesn't have much benefit to reuse the stale second-order information, which is an essential approach to reduce the frequency of evaluating the inverse and to make second-order methods practical.
- Applying M-FAC to fine-tuning a sparse model (less dimensionality) is indeed efficient.
  - However, as mentioned above, there is no convincing theoretical reason to achieve better test accuracy.

Other comments
- L67: “exactly computing the IHVP”
    - I feel that it is inappropriate to call it “IHVP” (in many parts of the paper)
    - What M-FAC calculates is inverse empirical Fisher vector products.
- L79: $\hat{F}_m^{-1}v$ -> $(\hat{F}_m+\lambda I)^{-1}v$

References:
1. Ashia C. Wilson et al.. The Marginal Value of Adaptive Gradient Methods in Machine Learning
, 2017. https://arxiv.org/abs/1705.08292.
2. Neha S. Wadia et al., Whitening and second order optimization both make information in the dataset unusable during training, and can reduce or prevent generalization, 2020. https://arxiv.org/abs/2008.07545.


=========================
Update after the discussion

Most of my concern came from my misunderstanding: (1) M-FAC cannot achieve fast convergence due to the limited sample size and the low-quality estimate of the curvature, and (2) the authors claim that M-FAC can somehow achieve a better test accuracy. But these points turned out not to be the case, and now I appreciate this paper's “optimization part” (dynamic algorithm) as well as the "pruning part" (static algorithm). Given the results in the "pruning part" and "optimization part," I am convinced that the techniques (i.e., static algorithm and dynamic algorithm) presented in this paper are indeed useful to evaluate the inverse-empirical-Fisher-vector product and interesting to the community. Therefore, I am happy to raise my score from 5 to 7 and recommend accepting.

**Time Spent Reviewing:**

5.5 hours

---

> ### Author Response · Authors · 2021-08-10
> **Response to Reviewer 3 JscG**
>
> Thank you for the detailed comments and suggestions!
>
> To our reading, your review raises three main negative points, which we address in turn, after which we discuss remaining questions. We would also be happy to further engage on any of those points during the discussion period.
>
> 1. _Lack of sufficient novelty_ (our work "does not contain enough ideas to be appreciated by the community, especially practitioners")
>
> Novelty is clearly subjective, but we have to say that we found this evaluation somewhat harsh.
> The static pruning algorithm is quite simple, but it appears to be new, and it can improve on the performance of the best previous approach (WoodFisher) by more than an order of magnitude in terms of performance (please see Table 2), and leads to highly-accurate pruned models (please see Table 1 and improved results below).
>
> The “online” approach used for the dynamic “optimization” algorithm is also new, and quite different relative to WoodFisher. (In particular, WoodFisher simply could not be used as a preconditioner for optimization, due to the massive per-step overheads described in Table 2.) When executed at comparable parameter values to other full-matrix preconditioners, M-FAC leads to significant performance improvements, and is quite competitive with other approaches in terms of accuracy, without employing momentum.
>
>
> Even if the reviewer were to find all of our applications completely unconvincing, we believe that highly-efficient algorithms for maintaining inverses under low-rank updates should be generally interesting for the community.
>
> 2. Lack of practical applications:
>
>
> a. In the context of pruning, M-FAC allows order-of-magnitude increases in the block size and number of gradients used to approximate the OBD update, without slowdown. These increases directly translate to improved accuracy for pruned models.
>
>
> We already provided evidence in Tables 1 and 2, where we only varied the number of gradients $m$ but kept other parameters constant for a direct comparison with WoodFisher (WF).
> To reinforce this point, for the rebuttal we have experimented with pruning schedules that perform more frequent pruning steps (and thus would be either very slow or impossible to run with WF) thereby achieving clear additional accuracy gains:
>
> - MBv1 @ 0.89 sparsity: 67.3 Top-1 (M-FAC) vs. 63.9 Top 1 (WF)
> - RN50 @ 0.98 sparsity: 66.9 Top-1 (M-FAC) vs. 65.6 Top-1 (WF)
>
> More precisely, we compute up to 4 times as many inverse Hessian estimations; nevertheless, in both cases, our schedules (using M-FAC) are still faster in terms of wall-clock time than the original WF schedules. We also note that the resulting sparse models provide inference speedups of up to 3x when executed on a sparsity-aware CPU inference engine (https://github.com/neuralmagic/deepsparse).
>
> Regarding the number of gradients $m$ used for M-FAC pruning, we note that moderate values such as the ones we use in the paper already work well (in addition, we use gradients averaged over small batches, e.g. size 32, rather than individual ones to reduce variance) and that we observe clear saturation effects when going far beyond those. Further, in the context of pruning larger-scale models, please note that the required storage (and compute) scales linearly with the current model sparsity and that M-FAC can also be applied independently for individual (or groups of) layers when pruning extremely large models.
>
> b. For optimization, we agree that the dynamic M-FAC algorithm is not suitable for training giant models, due to the gradient storage cost. However, training highly-accurate but small and/or sparse models to be used for inference on mobile/embedded devices with limited compute power is also an important research direction. For training such models, the memory required for gradient storage is not a limitation (especially with GPU memory steadily increasing from generation to generation) and this is thus an area where M-FAC (and follow-up work) could indeed be very effective, especially given that our initial results are already encouraging.
>
> 3. Lack of theoretical justification:
>
> We would first like to emphasize that the M-FAC optimizer is, unlike Adam, GGT or AdaHessian, _not_ an adaptive method. It is an approximation to natural (stochastic) gradient descent, like K-FAC. (For instance, please note that our preconditioning does not contain square roots.) Thus, the work of Ashia C. Wilson et al. on limitations of adaptive methods does not seem to directly apply.
>
> Regarding guarantees, under the approximations detailed in our paper, M-FAC has the same guarantees as K-FAC or other natural gradient approaches. The theory of natural SGD is well studied, from the early work of Amari [3] to the more recent https://arxiv.org/abs/1905.10961, which provided fast convergence rates for such algorithms in the context of neural networks. More precisely, our method mainly relies on the assumption that second-order information does not change too quickly during optimization. This property has been already validated in the K-FAC literature (e.g. https://jimmylba.github.io/papers/nsync.pdf); and we provide a clear justification for it in the context of our algorithm in Appendix D11.
>
> Second, regarding theoretical justification for _pruning_, we are implementing the (theoretical) OBD pruning framework in the context of deep neural networks.
>
> We will make these points more clear in the next revision.
>
> 4. We now proceed to address the other points raised:
>
> We also noticed that optimizers with rich second-order information typically learn quickly during the early epochs but then plateau at lower accuracies than first order methods. The faster initial learning (even in wallclock time) can also be observed for M-FAC: please refer to the wall-clock time results in Appendix D10. Interestingly, the plateauing you suggest does not seem to occur for M-FAC, which we consider a major advantage over other second-order methods. Again, we agree that this should be discussed in our next revision.
>
> > No error bars (by varying initialization points) are reported, so it is unclear whether M-FAC shows better accuracy by chance.
>
> As we state in Appendix D9, all experiments in Table 4 Left were repeated 5 times with different random seeds and we always report the median result. Our standard deviations were all quite small with values between 0.1 and 0.2. We agree that error bars / standard deviations should be included and we will add them in the next revision.
>
> > Due to the recursive nature of M-FAC (and the WSM inversion formula), it doesn't have much benefit to reuse the stale second-order information, which is an essential approach to reduce the frequency of evaluating the inverse and to make second-order methods practical.
>
> M-FAC can also be adapted to reuse stale second order information: for example, we can update the matrices $D$ and $B$ only every $k$ steps (replacing $k$ gradients simultaneously) to substantially decrease the amortized preconditioning overhead (e.g. to 1.15x for both RN20 and RN32 with $k = 8$) with only a small impact on the overall optimization performance (for example, we observed only ~0.25 lower accuracy on RN20). However, since M-FAC is already quite fast even without this additional approximation (which would also introduce another hyper-parameter that needs to be tuned), we focussed on the theoretically best approach of always using the most recent second-order information available. Nevertheless, we agree that it would be useful to point out this reuse possibility in the paper and will do so in the next revision.
>
> > Applying M-FAC to fine-tuning a sparse model (less dimensionality) is indeed efficient. However, as mentioned above, there is no convincing theoretical reason to achieve better test accuracy.
>
> Again, please notice that our algorithm has the same guarantees as natural gradient in this case as well.
>
> Thank you for your other comments, which we will carefully implement. In particular, we agree that "exactly computing the IHVP" is not precise enough and will fix the phrasing.

---

> > ### Comment · Reviewer_JscG · 2021-08-19
> > **I appreciate the "pruning part", but the "optimization part" is not convincing**
> >
> > Thank you for the clarification.
> >
> > First of all, I would like to clarify that I appreciate the “pruning part” (the static algorithm) in this paper and the authors’ response (regarding the number of gradients $m$).
> > M-FAC enables us to use a larger sample size $m$ than the previous WoodFisher algorithm, leading to a better accuracy-sparsity trade-off in model compression.
> > So it is clearly a novelty.
> > However, I do not think the results presented in the “optimization part” are convincing to show the effectiveness of M-FAC (the dynamic algorithm), as I will describe below.
> > Based on the results of the “pruning part” alone, I get the impression that M-FAC is WoodFisher’s “efficient implementation trick” (constructing/storing vectors instead of matrices).
> > The underlying mathematical ideas are not novel: calculate (layer-wise) inverse-empirical-Fisher-vector product by the Woodbury-Sherman-Morrison inversion formula.
> > Therefore, although I appreciate the trick's effectiveness, I still feel that the scientific contributions of this paper are not sufficient to be presented in NeurIPS.
> >
> > > The “online” approach used for the dynamic “optimization” algorithm is also new, and quite different relative to WoodFisher.  When executed at comparable parameter values to other full-matrix preconditioners, M-FAC leads to significant performance improvements, and is quite competitive with other approaches in terms of accuracy, without employing momentum.
> >
> > Yes, I agree that M-FAC is a high-performance full-matrix preconditioner *only when the sample size $m$ is small*.  In large-batch settings, where one can expect less biased and more stable curvature information, second-order methods by WSM inversion formula are infeasible and methods such as K-FAC are more suitable. So whether an algorithm is better than the others depends on the settings. M-FAC is feasible even with a large model while K-FAC is feasible even with a large batch. Comparing M-FAC and the other methods only in a small-batch setting seems not fair.
> >
> > > we believe that highly-efficient algorithms for maintaining inverses under low-rank updates should be generally interesting for the community.
> >
> > I agree with this statement. But an implementation trick (i.e., constructing/storing vectors instead of matrices, in this work) could be presented in a blog post (e.g., a trick for calculating Jacobian vector products by reverse-mode auto-diff https://j-towns.github.io/2017/06/12/A-new-trick.html) to help the community.
> >
> > > training highly-accurate but small and/or sparse models to be used for inference on mobile/embedded devices with limited compute power is also an important research direction. For training such models, the memory required for gradient storage is not a limitation (especially with GPU memory steadily increasing from generation to generation)
> >
> > I agree with this statement. And M-FAC (and second-order methods by WSM inversion formula in general) is indeed easy-to-evaluate in training small and/or sparse models.
> >
> > > this is thus an area where M-FAC (and follow-up work) could indeed be very effective especially given that our initial results are already encouraging.
> >
> > However, I do not see the results as “encouraging”. There is no convincing reason why M-FAC could achieve a better test accuracy in general because it is an “efficient implementation trick” of (layer-wise) stochastic NGD with full empirical Fisher, with which one can expect a faster convergence but a better test accuracy. Moreover, what the authors present is the result of “fine-tuning a sparse model”, not “training a sparse model from scratch”. For the former, where training time is inherently short, it is questionable whether even fast convergence (by a complex implementation) is necessary.
> >
> > > We also noticed that optimizers with rich second-order information typically learn quickly during the early epochs but then plateau at lower accuracies than first order methods. The faster initial learning (even in wallclock time) can also be observed for M-FAC. Interestingly, the plateauing you suggest does not seem to occur for M-FAC, which we consider a major advantage over other second-order methods.
> >
> > This phenomenon is indeed interesting. But again, there is no convincing reason why this (higher test accuracy) could happen in general.
> > I believe the potential advantage of M-FAC is a faster convergence due to the more accurate empirical Fisher.
> > However, in Figure 1 (c) (Epoch vs Training loss), M-FAC is as fast as the other second-order methods and is even slower than SGD and Adam. I would expect a faster convergence with a more aggressive (larger) learning rate for second-order methods. If this is not the case, it may be due to the small number of samples and the low quality of the curvature matrix estimation (the limitation of M-FAC and second-order methods by WSM inversion formula in general).
> >
> > > The theory of natural SGD is well studied … This property has been already validated in the K-FAC literature … and we provide a clear justification for it in the context of our algorithm
> >
> > Thank you for pointing these out, but these do not ensure a solution with a better test accuracy (which the authors present as the advantage of M-FAC).
> >
> > >  our algorithm has the same guarantees as natural gradient in this case as well.
> >
> > First, the guarantees are not the “same” as natural gradient as M-FAC uses empirical Fisher (this is not the point, though).
> > But yes, so there is no guarantee of better test accuracy.
> > As I mentioned in my review, we have an opposite insight (as the authors pointed out, the reference [1] was not appropriate for this claim).
> > Hence, even if M-FAC shows better test accuracy in the experiments, I am not convinced that it works in general.
> >
> > Finally, for the improvement of the "optimization part" of this work, I would suggest exploring whether M-FAC (with a limited sample size $m$) can achieve (i) fast convergence, (ii) fast training time, and (iii) good test accuracy by combining with some regularization tricks (or some theoretical justification if the authors believe that M-FAC generalizes well even without any reuglarization tricks).
> > Otherwise, I feel it is difficult to overcome the "intrinsic limitations" (i.e. a limited sample size $m$) of M-FAC (and methods based on the WSM inversion formula in general).

---

> > > ### Author Response · Authors · 2021-08-20
> > > **Reviewer Response Part 1: Clarification of Main Concerns**
> > >
> > > Many thanks for engaging with us, and for the opportunity to have a discussion.
> > >
> > > We start by clarifying what may be a significant misunderstanding regarding the behaviour of our optimizer, and in particular the relationship between train and test accuracy.
> > >
> > > To our reading, it appears that the reviewer is taking the results of Figure 1 (c) in our submission to represent a comparison of the "optimization speed" of each method. Thus, it may seem that we are claiming that, although our optimizer doesn't optimize training loss "as fast" as other methods (notably Adam), it somehow manages to have good test accuracy, without further justification.
> > >
> > > This is not the case: as we have stated in the submission text, and clarified in the response to Reviewer 1 (YzjZ), Figures 1 (c) and (d) execute each optimizer *under its own best specific hyper-parameters in terms of TEST accuracy*. Thus, the curves *cannot* be used to draw conclusions regarding the speed of loss optimization: Losses are not comparable in this figure, since
> > > 1.  the *loss structure* in this experiment is different for each method: due to different values of weight decay/L2 regularization, some methods incur higher training loss to reach their best test accuracy
> > > 2.  methods such as K-FAC and GGT require specific learning rate values and schedules for best results, so optimization trajectory is impossible to compare directly.
> > >
> > > We have presented things this way to provide a fair comparison of the main relevant metric for this experiment, which was *test* accuracy. We realize now that the *loss* graph in Figure 1 (c) may lead to wrong conclusions, and will revise the presentation of these results.
> > >
> > > We now wish to clarify the issue of convergence speed of the M-FAC optimizer, and will touch upon approximation quality w.r.t $m$, as well as generalization in practice.
> > >
> > > On convergence speed, please note that Appendix D.10 already presents accuracy vs. wallclock time plots for all the dense experiments in Table 4 (right). These show that M-FAC optimizes fastest initially (as expected). Second, Figure 10(b) in the Appendix shows the improved approximation quality of M-FAC relative to K-FAC (in the context of pruning, but the results apply more broadly).
> > >
> > > We further showcase convergence speed via the following experiment:
> > > We re-ran the WRN-40-2/CIFAR100 experiment in the setup from Table 4 (right), with a training schedule that is _compressed by 4x_ in terms of epochs, but keeping all other parameters the same. We plot training loss / accuracy vs. number of epochs at this anonymized link:
> > >
> > > https://github.com/M-FAC/M-FAC_neurips21/blob/main/short_training_WRN402_CIFAR100.pdf
> > >
> > > These plots clearly show the faster loss convergence of M-FAC, as expected. Importantly, this happens both in terms of loss and in terms of test accuracy, and other WRN models have the same behaviour. This suggests that the M-FAC approximation is already useful for moderate values of $m$: we used $m = 1024$ here, as well as in most other experiments.
> > > Moreover,  notice that SGD *without* momentum performs poorly in this experiment; M-FAC's improvement relative to this comes solely from preconditioning (we do not use momentum). Lastly, we want to mention that the final test accuracy of M-FAC in this 50 epoch run (70.4) actually exceeds the final test accuracy (69.7) of momentum SGD when run for 100 epochs (i.e. a 2x compressed schedule).
> > >
> > > At this point, we also want to briefly clarify that the input to M-FAC is *not* individual sample gradients, but rather *batch* gradients over the last $m$ steps, which are less noisy.
> > > For example, in our ImageNet experiments, this means that, with $m = 1024$ and batch-size 256, an M-FAC step integrates ~250K individual samples, which amounts to > 20% of the whole training set. (Although some of these gradient samples are averaged as part of a batch.)
> > >
> > > Finally, we discuss the connection between training accuracy and generalization in practice.
> > > We emphasize that, across all our experiments, M-FAC achieving better test accuracy is strongly correlated to it achieving lower training loss, as long as the objectives are comparable.
> > >
> > > For example, here are the final loss values corresponding to experiments in Table 4 Right:
> > > MBv1 dense  - 0.74 (M-FAC) vs. 0.79 (SGD),
> > > MBv1 sparse - 1.75 (M-FAC) vs. 1.91 (SGD),
> > > RN50 sparse - 1.21 (M-FAC) vs  1.43 (SGD).
> > >
> > > The same is observed in our language modelling results with BERT models (please see our discussion with Reviewer 2 6ujH), available here:
> > > https://github.com/M-FAC/M-FAC_neurips21/blob/main/Transformer_with_train_loss.pdf
> > >
> > > In summary, these results show that M-FAC’s good test accuracies are primarily due to the better optimization performance (which in turn is a result of the improved inverse Hessian approximation) and *not* due to any implicit regularization mechanisms (which we never claimed). Moreover, our method is fully compatible with common regularization approaches, in particular weight-decay or data augmentations.
> > > We agree with the reviewer that the presentation can be improved to clarify this, we will include both loss numbers and additional discussion in the next revision.
> > >
> > > We now list the reviewer’s remaining main concerns, and address them in detail.
> > >
> > > **Concern 1: Lack of theoretical justification for the optimization results:**
> > >
> > > The above clarification should partly address this concern, as our improvements in terms of test error are clearly correlated with lower training loss.
> > > In addition, the improved generalization accuracy can be explained theoretically by the fact that the M-FAC optimizer is a faithful implementation of Amari's efficient natural gradient [3], but at the scale of deep neural networks. Specifically, in [3], Amari also proposes the use of WSM to approximate natural gradient; in [3, Section 4], he proves that the resulting classifiers are "Fisher efficient," i.e., that they provide asymptotically optimal estimators of the underlying distribution, in the statistical sense.
> > > Relative to [3], our only additional assumptions are that
> > >  (1) The Empirical Fisher can be well-approximated via gradient sampling;
> > >  (2) Delayed gradients can be used instead of "fresh" gradients without significant distortion.
> > >  Both assumptions are common in practice, and we provide new evidence for both in the paper. Namely, Assumption (1) is justified by e.g. our one-shot pruning experiments, which show accuracy saturation while increasing the number of gradients.
> > > Assumption (2) is backed by the experiments in Appendix D.11 (showing high correlation in terms of cosine similarities of “stale” and “freshly computed” directions).
> > > Therefore, under these assumptions, the arguments of [3] apply, which would at least in part justify our good generalization performance.
> > >
> > > Clearly, this argument is not formal, and we do not have a *theoretical* justification for why our optimizer behaves better. Yet, we are unaware of full formal justifications why even SGD provides better *generalization accuracy* relative to e.g. vanilla gradient descent (GD) in the case of neural networks, although this is clearly the case. Intuitive interpretations exist ("flat minima"), and there are non-trivial analyses in toy cases. Asking us to prove formally that this is indeed the case seems like a very high bar, and one that very few methods, if any, would be able to clear. Please let us know if we are misunderstanding.
> > >
> > > **Concern 2: "optimization results are not enough to show the effectiveness of M-FAC"**
> > >
> > > Please recall that we perform three three types of optimization experiments:
> > >
> > >    a. **"Tuned" comparison with other methods on image classification tasks** (ResNet20, ResNet32 on CIFAR-10). (Figure 1 (c-d), with results in Table 4 (left).)
> > >    *Here, tuning is performed independently for each method. In addition to Appendix D.9, we have also further detailed our tuning process in the responses to Reviewers 2 (6ujH) and 4 (zd2D).*
> > >
> > >    These experiments essentially show that there exist parameter values for which M-FAC essentially matches or outperforms all other methods, when tuning is applied. The closest method in terms of accuracy (AdaHessian) is at least 2x slower in wall-clock time relative to M-FAC.
> > >
> > >    b. **Suitability as a "drop-in" replacement for Adam/SGD.** For this we trained 1)  WideResNets and MobileNet on CIFAR-100 and ImageNet, and BERT models on language modelling tasks. (Results in Figure 4 (right), Appendix D.10, and the response to Reviewer 2 (6ujH) for BERT --- please see link above.)
> > >    *Here, we simply ran M-FAC with default parameters in the standard learning setup for Adam or SGD.*
> > >
> > >    In brief, results show that M-FAC consistently outperforms Adam (e.g. improves upon Adam's accuracy significantly on 12 out of 14 BERT language modelling experiments), and can match and even outperform tuned SGD accuracy (e.g. we improve upon the Torchvision model accuracy for ResNet18/ImageNet, which we presume is well-tuned). This occurs both with and without weight decay, showing that our method is indeed compatible with regularization.
> > >
> > >    c. **Fine-tuning sparse models** (ResNet50/ImageNet and MobileNet/ImageNet, versus SGD, Table 2 right, bottom).
> > >    *Here, we again directly optimize under the vanilla SGD hyper-parameters.*
> > >
> > > Results show a clear accuracy difference in this case, as the reviewer also agreed.
> > >
> > > In summary, these results lead us to believe that our method can be practically useful.  Specifically, in the case of BERT optimization, they improve upon the accuracy of carefully-tuned mobile models trained by Google and HuggingFace, which are quite popular: the BERT-tiny model has been downloaded ~85'000 times over the last month, and M-FAC training improves its F1-score by > 4% relative to the HuggingFace Adam baseline, on SQUAD question-answering.
> > > This suggests that M-FAC can serve as a drop-in replacement for Adam, as it consistently provides higher accuracy at the same parameter values.

---

> > > > ### Comment · Reviewer_JscG · 2021-08-23
> > > > **The "optimization part" turned out to be convincing, and I would like to raise my score**
> > > >
> > > > Thank you again for the clarification.
> > > >
> > > > > it may seem that we are claiming that, although our optimizer doesn't optimize training loss "as fast" as other methods (notably Adam), it somehow manages to have good test accuracy, without further justification.
> > > >
> > > > Yes, this was exactly my take. And thank you for clarifying that “improvements in terms of test error are clearly correlated with lower training loss” with some convincing results.
> > > > Most of my concern came from my misunderstanding: (1) M-FAC cannot achieve fast convergence due to the limited sample size $m$ and the low-quality estimate of the curvature, and (2) the authors claim that M-FAC can somehow achieve a better test accuracy. But these points turned out not to be the case, and now I appreciate this paper's “optimization part” (dynamic algorithm) as well. Given the results in the "pruning part" and "optimization part," I am convinced that the techniques (i.e., static algorithm and dynamic algorithm) presented in this paper are indeed useful to evaluate the inverse-empirical-Fisher-vector product and interesting to the community. Therefore, I am happy to raise my score from 5 to 7 and recommend accepting.
> > > >
> > > > I believe that the presentation can be improved by clarifying in the main text that (a) M-FAC can achieve faster convergence, (b) the improved test error is correlated with lower training loss, and (c) the number of samples M-FAC can integrate is not small due to the use of batched gradients.

---

> > > > > ### Author Response · Authors · 2021-08-24
> > > > > **Thank you!**
> > > > >
> > > > > Thank you again for the detailed discussion and helpful suggestions!
> > > > >
> > > > > We are very glad that we were able to resolve your questions in a conclusive manner, and will be
> > > > > implementing your recommendations. Specifically, we will clarify the convergence speed, the use of batch-gradients, the correlation between loss and test accuracy, as well as the connection to natural SGD. All these points will be integrated into the next revision.

---

> > > ### Author Response · Authors · 2021-08-20
> > > **Reviewer Response Part 2: Detailed Answers to Reviewer Questions**
> > >
> > > EDIT: For some reason our responses appear in the wrong order, please start with the first part below
> > >
> > > https://openreview.net/forum?id=EEq6YUrDyfO&noteId=8PvYGL3MKau
> > >
> > >    We now provide detailed answers to your points, and try to clarify some misunderstandings.
> > >
> > >    > In large-batch settings,  second-order methods by WSM inversion formula are infeasible and methods such as K-FAC are more suitable.
> > >
> > > We respectfully do not agree on this point, as M-FAC is in fact applicable to a large-batch setting.
> > >
> > > Specifically, assume a standard large-batch setup with several “worker” nodes, each generating individual gradients, which are then aggregated. The same could be done in M-FAC: the workers’ small-batch gradients can be 1) aggregated to produce a “large-batch” gradient, but also 2) integrated *individually* into the Fisher approximation at a step. As noted in our first response, our method allows “block gradient updates”, so this can be done efficiently, and will decrease the single-node storage cost.
> > > Moreover, we believe that M-FAC can be more stable to large-batch gradients than SGD. We provide preliminary evidence for this, by means of an ResNet20/CIFAR10 experiment where we examine momentum SGD and M-FAC test accuracies as we simply increase the batch size, leaving all other parameters constant at their standard values:
> > >
> > > https://github.com/M-FAC/M-FAC_neurips21/blob/main/increasing_batchSize_table.pdf
> > >
> > > Further, please note that there is recent work which questions the large-batch efficiency of K-FAC relative to SGD approaches: https://arxiv.org/pdf/1903.06237.pdf
> > >
> > >    >  So whether an algorithm is better than the others depends on the settings.
> > >
> > >    We completely agree, and we did not assert any "dominance" claims in terms of algorithms and methods. We do believe that our algorithm does have an "interesting" set of features in this context, and that it could be useful to practitioners in some settings.
> > >
> > > We also emphasize that we have clearly not exhausted all the possible optimizations for our approach: our baseline implementation can still be optimized for lower overheads, M-FAC could also be applied layer-wise for large models, and the basic dynamic algorithm could be parallelized so its computation occurs among more than one GPU.
> > >
> > >    > Comparing M-FAC and the other methods only in a small-batch setting seems not fair.
> > >
> > >    We stress that the results shown for K-FAC are at its *optimal batch size*, following its author’s specific fine-grained tuning for the corresponding experiment (https://github.com/tensorflow/kfac/blob/master/kfac/examples/keras/KFAC_vs_Adam_Experiment.md). We have applied a similar tuning process for each algorithm in turn for the first set of "tuned" experiments. We therefore believe our comparison is fair.
> > >
> > >    > But an implementation trick (i.e., constructing/storing vectors instead of matrices, in this work) could be presented in a blog post
> > >
> > > We respectfully disagree that our work could be confined to a blog post.
> > >
> > > To illustrate, the blog post cited by the reviewer shows a smart, but simple, trick to avoid complex custom code. By contrast, our dynamic algorithm is based on a new decomposition formula (Theorem 1); we found the existence of this type of decomposition surprising, as it is a new result in a very well-researched area. We also believe the reviewer may also be underestimating the complexity of mapping this theoretical formula onto an efficient algorithmic implementation. We would be very happy if the reviewer would examine Appendices A through C for evidence of the complexity of our algorithms and their implementations.
> > >
> > > > M-FAC is an “efficient implementation trick” of (layer-wise) stochastic NGD  with which one can expect a faster convergence but [not] a better test accuracy.
> > >
> > > We believe there may be some confusion here.
> > > First, the M-FAC optimizer is *not layer-wise:* the preconditioning is always computed *globally*. (Our block-wise algorithm is only used in  the *pruning* experiments. It is true that blocking layer-wise would be possible and would reduce overheads, but we chose not to do so, as it would add another approximation.)
> > > Second, as stated above, NGD _does_ have non-trivial statistical guarantees, which extend to M-FAC, under the corresponding assumptions, which we do justify in the paper.
> > >
> > > > Moreover, what the authors present is the result of “fine-tuning a sparse model”, not “training a sparse model from scratch”.
> > >
> > > Fine-tuning is part of the regular pruning process for most pruning methods, as they are usually obtained based on pre-trained *dense* models. We could try to train a sparse model from scratch, but 1) it's not clear how we would decide on the pruning mask, and 2) this usually results in worse accuracy of the pruned model than pruning and then fine-tuning a pre-trained one.
> > >
> > > > For the former, where training time is inherently short, it is questionable whether even fast convergence (by a complex implementation) is necessary.
> > >
> > > Indeed, the key metric for compressed models is *generalization accuracy*, not training time, and we did show that our method provides clearly higher accuracies in this context. Sparse training is a harder optimization problem (see e.g. https://arxiv.org/abs/1906.10732 for evidence), so a better optimizer is indeed expected to do better in this setting.
> > >
> > > However, we emphasize that training times for sparse networks can be quite long: STR and WoodFisher each fine-tune their models for around 100 epochs on ImageNet, and RigL (https://arxiv.org/abs/1911.11134) fine-tunes for up to 500 epochs. We showed that M-FAC can reduce training time from 100 to 30 epochs, with negligible loss of accuracy vs. SGD, in the same setting as STR/WoodFisher.
> > >
> > > > I believe the potential advantage of M-FAC is a faster convergence due to the more accurate empirical Fisher. However, in Figure 1 (c) (Epoch vs Training loss), M-FAC is as fast as the other second-order methods and is even slower than SGD and Adam. I would expect a faster convergence with a more aggressive (larger) learning rate for second-order methods. If this is not the case, it may be due to the small number of samples and the low quality of the curvature matrix estimation (the limitation of M-FAC and second-order methods by WSM inversion formula in general).
> > >
> > > We essentially agree on the first point, which is also supported by experiments: please see the ample clarification on this point provided at the beginning of this response. In particular, M-FAC is not slower than SGD / Adam, and *does* have faster convergence due to preconditioning, as the reviewer intuited. As shown in Appendix Figure 10 (b), M-FAC does provide a better local Hessian approximation than K-FAC.
> > >
> > > > As I mentioned in my review, we have an opposite insight (as the authors pointed out, the reference [1] was not appropriate for this claim). Hence, even if M-FAC shows better test accuracy in the experiments, I am not convinced that it works in general.
> > >
> > > Please note the clarifications regarding test vs train accuracy. We would really appreciate it if the reviewer could clarify their insight in more detail, so we could attempt to address it.
> > >
> > > > Finally, for the improvement of the "optimization part" of this work, I would suggest exploring whether M-FAC (with a limited sample size m) can achieve (i) fast convergence, (ii) fast training time, and (iii) good test accuracy by combining with some regularization tricks (or some theoretical justification if the authors believe that M-FAC generalizes well even without any regularization tricks). Otherwise, I feel it is difficult to overcome the "intrinsic limitations" (i.e. a limited sample size m) of M-FAC (and methods based on the WSM inversion formula in general).
> > >
> > > We thank the reviewer for the improvement suggestions, but would respectfully note that our proposal already has most of these features.
> > >
> > > Specifically, as discussed, M-FAC has lower training loss and better test accuracy for the same number of iterations relative to first-order methods, and the resulting models generalize well on all the task/model combinations we tried. The optimizer is fully-compatible with “regularization tricks,” and is shown to work as a drop-in replacement to Adam or SGD for BERT-type models and CNNs on ImageNet. When used with delayed updates, the optimizer has similar computational overheads to Adam on small models (15%), but consistently better accuracy.
> > >
> > > We would again like to thank the reviewer for engaging with us and stating their concerns, and hope that we addressed them in full. We would be very happy to keep the discussion going and clarify any outstanding issues.

---

### Official Review · Reviewer_6ujH · 2021-07-16

**Rating:** 7
**Confidence:** 4

**Summary:**

In this paper the authors derive a routine for calculating inverse fisher vector products, mostly based on the Woodbury-Sherman-Morrison Trick. They then demonstrate this routine by using it to compress and optimize neural network models.

**Limitations And Societal Impact:**

I agree with the authors that I do not believe there are relevant societal impacts of this work that are useful to discuss at this time.

**Main Review:**

After reviewing the answers to my review and seeing the author commitments to including additional details and Transformer experiments, I am raising my score from a 6 to a 7.

--------------------------------------------------

Overall I believe this is a useful and interesting paper. It is well written and sufficiently novel, with comparisons to a few appropriate prior works. Hopefully, with useful implementations and more thorough comparisons, it could be a useful deep learning optimizer and compressor. For example, the paper would be made much stronger by including any sort of model architecture besides a CNN on images, such as a Transformer based model, or better yet a model which one believes could particularly benefit from additional optimization preconditioning. I am willing to change my score to a more decisive value after a discussion with the authors!

Concerns:
Most of my concerns are regarding the optimization section. In general, for the optimization experiments, is your goal to demonstrate that MFAC improves final test accuracies, or training speed? Given that this is a preconditioner, I would think training speed would be a more useful metric to show, therefore it is more useful to also include the wallclock time it takes to reach some target test accuracy (for example, show the number of seconds it takes to reach 92% CIFAR test accuracy). It’s very useful to clearly state what it means to “improve” upon baselines in optimization, as some training algorithms tune/optimize for regularization, training speed, final test accuracy, or other different objectives. Some other questions:
-The appendix has good details for tuning the optimizers! One question that I still have is, were a similar number of hparam trials used for each method? I think it’s fine (and actually even a better idea) to use more tuning compute on baselines compared to a proposed method, so any extra tuning of GGT/KFAC is fine.
-It would also be useful to report the exact tuning ranges/grids used.
-What damping values were used for GGT/KFAC/MFAC? If one is much larger than others then that could be more like sgd. You say you used “an initial dampening of ≈ 0.1887” for KFAC, did you use a damping decay schedule or keep it fixed?
-Fig 1d would be much more readable if zoomed in more (showing only accuracies above ~60%?)
-Fig 1d seems to imply that an LR decay drop at 50/75% of training is very beneficial to the methods that used it; would it be possible to include one in the GGT/KFAC LR schedules? You could either reuse the same schedule as GGT/KFAC and add the decays in, or try more tuning on the initial LR but keeping the other hyperparameter adjustments?

Correctness:
-For the TF KFAC baseline, did you set invert_every=1 if using kfac.PeriodicInvCovUpdateKfacOpt? If not, how did you set up the frequency of the covariance/inversion ops?
-Did you use a validation set for the CIFAR experiments? For ImageNet it is unfortunately more common to tune on the test set because everyone uses the validation set as the test set (but I believe one should still carve out a slice of a training set), but for CIFAR there is a common convention to use 10% of the CIFAR10 training set as a validation set. Of course, you can always retrain on the full training set after selecting hyperparameters! While this may not change the ranking of baselines because (presumably) the same tuning budget and the same number of hyperparameters were used for each optimizer.
-Was weight decay not used with Adam (no value was listed in Table 6 in the appendix. If not it would be useful to include it as another baseline for a fairer comparison, especially on CIFAR where overfitting is a big concern.

Writing:
Nit: (8) and (11) are both missing a \nabla in front of \ell

Additional feedback:
-Table 2 is a great direct comparison.
-It’s great to see the provided CUDA kernel code.
-Appendix D.11 is very interesting, it would be useful to reference it from the main text!
-I’m surprised that you don’t need momentum to get MFAC to be competitive, it isn’t necessary but adding momentum would be an interesting addition

**Time Spent Reviewing:**

5

---

> ### Author Response · Authors · 2021-08-10
> **Response to Reviewer 2 6ujH**
>
> Thank you for your detailed comments and suggestions!
>
> > For example, the paper would be made much stronger by including any sort of model architecture besides a CNN on images, such as a Transformer based model.
>
> Following the reviewers suggestions, we tested the M-FAC optimizer on Transformer models for several NLP tasks. This appears to be a promising application, as our results usually improve upon Adam variants, the optimizer of choice for this type of model.
>
> Due to time constraints, we focused on finetuning of the BERT-tiny (4.3 million parameters) and BERT-mini (11.2 million parameters) variants of the well-known BERT model on language modelling tasks: a question answering task (SQUADv2) and on several text classification tasks contained in the GLUE benchmark suite (we only exclude RTE and WNLI datasets due to the low number of data samples).
> We integrated M-FAC into the widely used HuggingFace implementations (https://github.com/huggingface/transformers/tree/master/examples/pytorch), adopt their suggested training setup (e.g. batch-size, number of epochs, learning rate schedule, etc.), and compare with their tuned Adam baselines. For M-FAC, we always use learning rate 1e-4 and dampening 1e-6 (we found that finetuning Transformers requires slightly lower values, so we dropped our 1e-3/1e-5 CNN defaults both by a factor of 10).  We use $m = 1024$ gradients for M-FAC, but reduce this for tasks where significantly less than 1000 overall training gradients are available. Neither the Adam baseline nor M-FAC use weight decay. We repeat all experiments 5 times and report mean and standard deviation. All results are presented in the following tables that due to their size are linked here (in an anonymous repository):
>
> https://github.com/M-FAC/M-FAC_neurips21/blob/main/Transformer_model_results.pdf (Section 1 & 2)
>
> We can see that M-FAC outperforms the Adam baseline on SQUADv2 as well as on the large majority of GLUE tasks. We note that this is despite using default parameters for M-FAC as well as fully adopting the training setup that was designed for Adam and thus may not be ideal for M-FAC.
>
> Additionally, we reran our BERT-tiny experiments with 4 epochs of finetuning, submitted them to the GLUE test servers (https://gluebenchmark.com/) and compare the results with the BERT-tiny authors’ competitive baselines
> (https://github.com/google-research/bert) which are the result of a hyper-parameter search and use AdamW (they also finetune for 4 epochs). Already without any tuning, M-FAC performs on par or better than these baselines for
> several tasks. Further, we find that for datasets with a very low number of samples (i.e. gradients), a modest tuning of dampening and learning rate helps. Our results are presented in Section 3 of the table document linked above.
>
> All in all, while we are sure that most results presented here can be improved with more careful parameter tuning (in particular, by applying weight decay to M-FAC), we believe that this is already a very promising start for the applicability of M-FAC to Transformer models and think that a more thorough exploration will be a very interesting topic for future work.
>
> > What do we want to show with the optimization experiments?
>
> Our goal was to show that M-FAC is an optimizer with rich second-order information that is able to achieve competitive accuracies while running in standard optimization setups (e.g. batch-size, simple learning rate schedule), with only moderate amounts of tuning (we only tune weight decay values and keep dampening and learning rate constant) and having significantly lower wall-clock overheads compared to other second-order optimizers running in a comparable setup.
>
> > Hyper-parameter search details
>
> In general, our hyper-parameters for the experiments in Table 4 Left are a combination of proven values found in literature and the result of our own small grid-searches. In particular, the SGD learning rate & momentum are from [13] and learning rate & momentums for Adam, AdaHessian and AdamW (we discuss this new addition below) are from [39]; all of these values were tuned by the respective works. The M-FAC default dampening 1e-5 and learning rate 1e-3 were determined by small amounts of experimentation during development. Since we found that the exact weight decay value can have a significant impact on the final test accuracies, we performed grid-searches over commonly used values {0, 1e-5, 1e-4, 5e-4, 1e-3, 3e-3, 1e-2} for all methods that use weight decay. We note that this identified better values for AdaHessian and AdamW than the ones used in [39].
>
> Hyperparameter tuning for K-FAC:
>
> For experiments with the K-FAC optimizer on ResNet20/CIFAR10, we used exactly the set of heavily tuned hyperparameters provided by the authors at https://github.com/tensorflow/kfac/blob/master/kfac/examples/keras/KFAC_vs_Adam_Experiment.md with one change, that is setting `invert_every=1` for a fair comparison with other optimizers and because it improved accuracy noticeably by 0.4 percent. In fact, we used this code to run all our K-FAC experiments so momentum and exponential decays of learning rate and damping were also applied. For ResNet32/CIFAR10, the authors do not provide tuned parameters and the ResNet20 ones do not perform well. Therefore, we performed a search for better values over the following grid (the momentum and exponential decays are preserved from the ResNet20 settings):
>
>
> - Initial learning rate = {0.5, 0.25, 0.1, 0.05, 0.025, 0.01, 0.005, 0.0025, 0.001}
> - Final learning rate = {1e-4, 1e-5}
> - Initial damping = {0.3887, 0.2887, 0.1887}
> - Invert_every = {1, 10}
>
> Hyperparameter tuning for GGT:
>
>
> As noted in the paper, we use the GGT authors’ original implementation contributed to Tensorflow. Since there were no prior results for GGT on the models we consider, we contacted the authors for missing hyper-parameters, and we performed a search for the best set of hyperparameters over the following grid (we keep the various dampening parameters at their recommended defaults and use a cosine annealing schedule as recommended in the paper, see also our next answer):
>
>
> - Initial learning rate = {0.5, 0.1, 0.05, 0.01, 0.005, 0.001, 0.0005, 0.0001}
> - Window size = {50, 100, 200, 500}
>
> > Dampening for K-FAC / GGT / M-FAC:
>
> All M-FAC experiments use the constant dampening 1e-5 (we determined this as a reasonable default in the beginning and did _not_ retune it for individual experiments). K-FAC uses an initial dampening of ~0.29 (for RN20 and ~0.19 for RN32) that is decayed to 1e-6 with a rate of ~7e-4, which are exactly the tuned settings recommended by the authors in [1*]. GGT uses several forms of dampening (1e-4 for truncating small eigenvalues, 1e-6 for stabilizing the SVD and 1e-2 for regularizing the inversion), which we have also left at the default values recommended by the author’s implementation in [2*].
>
> > Learning rate drops
>
> All optimizers except for K-FAC and GGT use the same simple learning rate schedule with drops at 50% and 75% of training (suggested in the original ResNet paper [13]). We also tried to run K-FAC / GGT with this same schedule but got significantly worse results. We then followed the author’s recommendations of using exponential decay schedules (for K-FAC) / and cosine decay (for GGT) schedules. Thus, for fairness, we show results for the latter. Also, showing only accuracies above 60% in Figure 1d is a good suggestion, which we will implement in the next revision.
>
> Due to this complex process, all in all, we believe that we spent more time tuning these baselines than our own algorithm.
>
> > K-FAC `invert_every`
>
> As already mentioned, we set `invert_every=1` for a fair comparison with M-FAC, GGT and AdaHessian, which are all configured to update at every step (see also line 345 in the paper).
>
> > CIFAR10 validation set
>
> As discussed earlier in our response, we did use the same tuning budget for all but K-FAC / GGT, which used an increased budget (and for methods other than M-FAC, we bootstrapped some parameters with optimized choices from literature, so those may actually be tuned more carefully than M-FAC). Using a validation set is a good suggestion; we have done this for the rebuttal, and found the same weight decay values for M-FAC. We will note this in the next revision of the paper. (The results in Table 4 Right do not include any tuning for computational reasons but were based on reasonable default values determined from the experiments in Table 4 Left).
>
> > Adam weight decay
>
> Our Adam results are indeed for vanilla Adam without weight decay. We have extended our results with AdamW and will integrate those into the next revision:
>
> - AdamW: 91.78 (RN20), 92.58 (RN32)
>
> (learning rate = 0.01 from [39] and weight decay = 0.01 determined by our hyper-parameter search as it performed better than the value used by [39])
>
> > M-FAC and momentum
>
> We also find it very interesting that M-FAC does not seem to require any momentum, which is typically essential for good performance with first-order optimizers and also helps significantly for other second-order ones. Using momentum did not show any noticeable improvements in preliminary experiments, so we decided to run M-FAC without momentum. We speculate that M-FAC not needing momentum may be related to the fact that, as shown by the dynamic algorithm, preconditioned gradients are ultimately formed as linear combinations of the gradient window (with coefficients determined by our derived formulas). This might implicitly perform some kind of momentum. However, we think that a more careful investigation of the interaction between M-FAC and momentum is an interesting direction for future work.
>
> [1*] https://github.com/tensorflow/kfac/blob/master/kfac/examples/keras/KFAC_vs_Adam_on_CIFAR10.ipynb
>
> [2*] https://github.com/tensorflow/tensorflow/blob/v1.15.0/tensorflow/contrib/opt/python/training/ggt.py

---

> > ### Comment · Reviewer_6ujH · 2021-08-22
> > **Response**
> >
> > Thank you very much for clarifying answers to all my questions! I believe that including these details and the additional Transformer results (preferably with additional tuning and details) in the final paper will be very useful for future readers.
> >
> > > What do we want to show with the optimization experiments?
> >
> > That makes sense you would like to show faster optimization. I believe you are close to having the sufficient details in the paper already, but a more explicit statement that this is the goal would be useful, as well as plots/tables showing the total wallclock time for each optimizer. Ideally you could calculate the wallclock time required to reach a target validation accuracy (88% or 89% accuracy on CIFAR10 with ResNet 20 or 32?), which could be done if you still have the training/validation curves from your Table 4 experiments.
> >
> > After considering the responses I will be raising my score to a 7.

---

> > > ### Author Response · Authors · 2021-08-24
> > > **Thank you!**
> > >
> > > Thank you very much for your response, and for your helpful suggestions and questions!
> > >
> > > We will integrate the new Transformer experiments (with details) into the next revision, clarify the scope of our optimizer using the additional data discussed, and update the experimental details, as per our initial reply.

---

### Official Review · Reviewer_YzjZ · 2021-07-18

**Rating:** 7
**Confidence:** 2

**Summary:**

In this work, the authors propose efficient techniques for computing inverse of a matrix that is represented as a sum of rank-1 matrices. The basic idea is to use recursively well-known Sherman-Morrison formula for computing inverse of rank-1 update. The authors show that their technique can be applied for computing inverse of the empirical estimate of the Fisher Matrix and thus can be used in optimization algorithms and for so-called prunning of a trained model.

**Limitations And Societal Impact:**

-

**Main Review:**

The paper is well-written and it is pleasant to read. Based on the Sherman-Morrison formula, the authors propose efficient update formulas that can be used for computing the inverse of a static matrix represented as a sum of 'm' rank-1 matrices in time O(d * m^2), where 'd' is the dimension of the problem. Hence, this technique can be efficient when the dimension of the problem is very big.

Moreover, the authors show how to update inverse matrix, replacing just one component in the sum of rank-1 matrices, which requires O(dm^2 + m^3) time on preprocessing and O(dm + m^3) for one update. Taking into account that 'm' (the batch size) can be chosen to be small, these estimates seem to be quite reasonable for using in high-dimensional setting.

The authors provide efficient implementation of these methods and show that it can be used for two problems: static estimation of inverse of Fisher matrix at the optimum for prunning a trained model, and dynamic estimation of inverse of Fisher matrix for its usage in (preconditioned)
Stochastic Gradient Descent.

I have only one major question with the experimental results on optimization algorithms:
It is not clear to me, whether the presented graphs and numerical results in the table demonstrate better performance of the new method (M-FAC). Indeed, in Figure 1(c), we see that the best performance has the Adam algorithm, in terms of the Training loss vs. Epochs. The performance of M-FAC is not significantly better than the perforamnces of K-FAC and SGD. In Figure 1(d) with the Test accuracy vs. Epochs we see the similar picture. Moreover, I am not sure that comparison only in terms of the number of Epochs is fair: the computational time among different methods can be quite different, and as we see in Table 4 the overhead of M-FAC is bigger than the overhead of K-FAC and significantly bigger than Adam/SGD. Hence, it looks to me that the Adam algorithm demonstrates the best performance in the provided experiments, from the optimization perspective.


Some minor remarks:

1. line 107: It is worth to mention that $p_{ \theta }(x, y)$ is density function. Also, it is confusing that the arguments (x, y) and \theta are swapped, comparinng with the joint distribution $P_{(x, y)}(\theta)$.

2. line 110: Hessian of 'theta' -> Hessian of 'L' at 'theta' ?

3. line 249: Theorem 1 - the inverse of F in the beginning is redundant.

4. line 342: 'overhead over SGD' - I think it is worth to add more description on this measure (is it computational time or arithmetical operations?)

**Time Spent Reviewing:**

3

---

> ### Author Response · Authors · 2021-08-10
> **Response to Reviewer 1 YzjZ**
>
> Thank you for the detailed comments and suggestions!
>
> You raise a good point about the performance of Adam; Figure 1 (c) indeed seems to suggest that Adam optimizes best in terms of training loss. However, this is due to the fact that Adam does not use weight decay, which leads to a lower loss value and makes optimization “easier”. Notice however, in Figure 1(d) and Table 4 Left, that Adam consequently reaches significantly lower test accuracy than the best-performing methods (which use weight decay). We agree that this is a bit confusing and therefore extended these experiments to contain AdamW (the proper weight decay version of Adam) as well as M-FAC when not using weight decay.
>
> These reach the following results:
>
> - Adam (no wd, paper):     89.67 (RN20), 90.56 (RN32)
> - AdamW (with wd, new):  91.78 (RN20), 92.58 (RN32)
> - M-FAC (no wd, new):      91.48 (RN20), 91.88 (RN32)
> - M-FAC (with wd, paper): 92.34 (RN20), 92.65 (RN32)
>
>
> Further, in the updated plots (see them in the anonymous repository here: https://github.com/M-FAC/M-FAC_neurips21/blob/main/updated_optimizer_loss_and_accuracy_plots.pdf), one can see that, as expected, AdamW optimizes the loss slower and M-FAC (without weight decay) faster than vanilla Adam.
>
> Regarding the optimizer speed, the overheads we report are all in per-epoch wall-clock time relative to SGD, where SGD time is always measured in the corresponding framework / environment (We will make this more clear as you suggest in remark 4).
>
> When comparing the M-FAC overheads to K-FAC/GGT, please note that the former uses an 8x larger batch-size and the latter a 5x smaller gradient window (this is also discussed in lines 348 - 353). In parentheses (in Table 4 Left), we provide overheads when running K-FAC/GGT in a comparable configuration (i.e. same batch-size / same gradient window size) to M-FAC; those are significantly higher than M-FAC. We also study test accuracy vs. wall-clock time in Appendix D10, where one can see that M-FAC often learns fastest in the early epochs and reaches better test accuracies with the same or only slightly longer wall-clock training time than other methods (and despite not using any optimization schedules tuned for that purpose). We agree that this should be referenced / discussed in the main text and will do so in the next revision.
>
> Thank you also for your other remarks, which we will implement carefully.

---

### Author Response · Authors · 2021-08-10
**Rebuttal Overview**

We want to thank all reviewers for their detailed comments and useful suggestions!

To our reading, the main points raised by the reviewers are summarized as follows:

1) Concerns about practicality and flexibility of the M-FAC algorithms, in particular the ability to delay updates (reviewers 3 JscG & 4 zd2D)
2) Improving the selection of models / datasets used in the optimization experiments (reviewers 2 6ujH & 4 zd2D)
3) Theoretical guarantees (reviewer 3 JscG)
4) Detailed questions about our experiments (reviewers 1 YzjZ, 2 6ujH & 4 zd2D)

We provide an overview of how we addressed these points (please see our individual replies for details):

1) We clear up several misunderstandings regarding M-FAC’s performance and flexibility; in particular, M-FAC _can_ perform delayed updates, which further reduces overheads. Moreover, we provide additional pruning results and discuss additional practical applications.
2) We provide additional optimization results for two Transformer models applied to various natural language modelling tasks. We also show gradual pruning results for an object detection model and give new optimization results for a larger image classification model.
3) We clarify that our optimizer is an instance of the natural gradient framework, whose theoretical guarantees are well studied in the literature, and discuss our extra assumptions.
4) We carefully answer all questions; in particular, about tuning details, the definition of overheads, fair comparison with K-FAC & GGT, the difference between M-FAC and WoodFisher and the performance of Adam/AdamW.

We look forward to an interesting discussion period and are happy to further engage on any of the issues raised!

---

### Author Response · Authors · 2021-08-18
**Rebuttal Follow-up**

Dear Reviewers and ACs,

As a week has passed since our response, we would like to follow-up on it. We believe our rebuttal does address the issues raised in a substantive manner (e.g. in the form of the new BERT / language modelling experiments, or the results for the version of the algorithm with delayed updates), and we would be very glad to further clarify these or any other issues the reviewers may have.

With best regards,

The authors

---

### Decision · Program_Chairs · 2021-09-27

**Decision:**

Accept (Poster)

**Comment:**

This paper uses a recursive Woodbury formula to maintain a low-rank approximation of the empirical Fisher information matrix, and uses this for parameter pruning and preconditioning in stochastic training. While this approach is a somewhat obvious application of the Woodbury formula, it appears to be the first time it has been seriously applied in the context of curvature matrix approximations for deep neural networks.

The reviewers found the paper to well-written and thorough, and the method to be a (potentially) useful tool. Therefore I'm recommending acceptance.

However, I agree with many of the concerns raised by the reviewers, but am less convinced by the rebuttals than they were. The main one for me is that this method will not scale to large networks due to the need to store m parameter-shaped vectors. Relatedly, I found the claim made by the paper that this method has storage costs "linear in the dimension of the model" to be highly misleading, since m is not really a constant (and is anyway quite large even if it were just a constant). In reality, the storage costs of this method are a lot higher than pretty much every other method compared to.

I also don't view the optimization experiments as meaningful, and largely agree with the points raised by Reviewer JscG. It is known that preconditioning doesn't really accelerate training of residual networks at small/medium batch sizes (see https://arxiv.org/abs/1907.04164). This means that all optimizers will perform similarly on such problems, and any small differences in test accuracy are incidental and have nothing to do with faster optimization. It would be much more interesting to try these methods on problems where stronger optimizers are already known to make a substantial difference (or on ResNets at very large batch sizes as in https://arxiv.org/abs/1811.12019). I would also recommend that the authors remove the graph comparing training performance for the different optimizers, since they aren't even optimizing the same objective function. (And even if they were, it's not really meaningful to compare training accuracy curves using hyperparameters that were tuned for best final test accuracy.)